# The timing of differentiation and potency of CD8 effector function is set by RNA binding proteins

Georg Petkau[1], Twm J. Mitchell[1], Krishnendu Chakraborty[1], Sarah E. Bell [1], Vanessa D´Angeli[1], Louise Matheson [1], David J. Turner[1], Alexander Saveliev [1], Ozge Gizlenci [1], Fiamma Salerno [1], Peter D. Katsikis [2] & Martin Turner [1✉]

CD8[+] T cell differentiation into effector cells is initiated early after antigen encounter by signals from the T cell antigen receptor and costimulatory molecules. The molecular mechanisms that establish the timing and rate of differentiation however are not defined. Here we show that the RNA binding proteins (RBP) ZFP36 and ZFP36L1 limit the rate of differentiation of activated naïve CD8[+] T cells and the potency of the resulting cytotoxic lymphocytes. The RBP function in an early and short temporal window to enforce dependency on costimulation via CD28 for full T cell activation and effector differentiation by directly binding mRNA of NF-κB, Irf8 and Notch1 transcription factors and cytokines, including Il2. Their absence in T cells, or the adoptive transfer of small numbers of CD8[+] T cells lacking the RBP, promotes resilience to influenza A virus infection without immunopathology. These findings highlight ZFP36 and ZFP36L1 as nodes for the integration of the early T cell activation signals controlling the speed and quality of the CD8[+] T cell response.

[1] Immunology Programme, The Babraham Institute, Babraham Research Campus, Cambridge CB22 3AT, UK. [2] Department of Immunology, Erasmus University Medical Center, P.O. Box 2040, 3000CA Rotterdam, Netherlands. ✉email: martin.turner@babraham.ac.uk

CD8[+] T cells are instrumental for the clearance of pathogen infected or malignant cells and the provision of immune memory. Upon T cell receptor (TCR) sensing of peptide presented by MHC-I, naive CD8[+] T cells exit quiescence and engage a differentiation program to form cytotoxic T-lymphocytes (CTL) accompanied by massive clonal expansion. High affinity antigen overrides inhibitory mechanisms which ensure the quiescent state of the CD8[+] T cell and promotes growth and cell cycle entry[1,2]. Co-stimulatory signals, termed "signal 2", from CD28 are critical for lowering the activation threshold- especially when TCR stimulation is suboptimal[3].

The amount and duration of TCR stimulation correlates with the clonal expansion of antigen-specific cells[4]. However, weak TCR signals are sufficient to induce a full differentiation program, albeit more slowly[5,6]. The cytotoxic effector differentiation program is installed early after activation with estimates ranging between 2- and 48-h after antigen stimulation[4,7–10]. Persistent exposure to IL-2, IL-12 and type-I interferon further shapes the response of CD8[+] T cells by providing critical survival factors and promoting effector differentiation[11–13]. The convergence of these signals on the regulation of transcription and chromatin states has been clearly demonstrated[14–16]. By contrast, an understanding of the nature and influence of RNA binding proteins (RBP) on CD8[+] T cell activation and differentiation is very limited[17].

Amongst RBP, the ability of Roquin (Rc3h1) and Regnase1 (Rc3h12a) to limit CD8[+] T cell responses has been linked to the repression of TCR signaling and cytokine production[18,19]. Their absence in T cells leads to T cell hyperactivation and autoimmune/inflammatory disease. The ZFP36 family of RBP bind AU-rich elements (AREs) present in the 3′ untranslated region (3′ UTR) of mRNAs and can effect different outcomes promoting RNA decay[19], suppressing translation[17,20,21] or directing localised translation[22] which are cell-context-specific[23]. Of the three ZFP36 gene-family members expressed by CD8[+] T cells, ZFP36L2 is present in naive and memory cells, while ZFP36 and ZFP36L1 are induced rapidly and transiently following TCR stimulation[20]. Zfp36-deficient mice show heightened immune responses and develop a severe autoimmune syndrome attributable to its function in myeloid cells[19,20]. An enhanced CD8 response in Zfp36-deficient mice has been linked to the excessive production by myeloid cells of IL-27, of which the p28 subunit is a direct target of ZFP36[24]. In quiescent memory CD8[+] T cells the ZFP36-paralog ZFP36L2 suppresses the translation of cytokine mRNA[25], but no studies have yet investigated the biology of ZFP36L1 in T cells. The widespread expression of ZFP36-family members by haematopoietic and mesenchymal cells and the genetic redundancy between them has made it challenging to understand the contributions of these genes to T cell physiology.

In this study we show ZFP36 and ZFP36L1 function during an early temporal window after activation in CD8[+] T cells to limit the tempo of effector differentiation and the functional capacity of differentiated effector cells. CTL deficient for both ZFP36 and ZFP36L1 show superior cell intrinsic cytotoxicity and confer greater protection against influenza A virus infection. An important function of ZFP36 and ZFP36L1 is to suppress T cell activation and enforce dependence upon costimulatory signals via CD28. The RBP bind transcripts encoding subunits of the NF-κB pathway, Notch1, Irf8 and Il2 to regulate the abundance of their protein products early after T cell activation. Thus, the acquisition of the CD8 effector fate is under the dominant control of ZFP36 and ZFP36L1.

## Results

**ZFP36 and ZFP36L1 limit the anti-viral CD8[+] T cell effector response.** To examine the consequences of the absence of ZFP36 and ZFP36L1 in T cells during an immune response we infected control mice Zfp36[fl/fl]/Zfp36l1[fl/fl] (WT), Zfp36[fl/fl]CD4[cre] (ZFP36 KO), Zfp36l1[fl/fl]CD4[cre] (ZFP36L1 KO) or Zfp36[fl/fl]/Zfp36l1[fl/fl]CD4[cre] (dKO) mice intranasally with a sublethal dose of H1N1 influenza A virus A/Puerto Rico/8/1934 (IAV/PR8). Following infection, using body weight as a surrogate for morbidity, we observe no significant difference between WT mice and mice with T cells deficient for ZFP36 or ZFP36L1 (Fig. 1a, b). By contrast, dKO mice show significantly less weight loss when compared to WT mice (Fig. 1c). Moreover, when challenged with a lethal dose of IAV/PR8 dKO mice show increased survival compared to WT mice (Fig. 1d). We conclude that the absence of ZFP36 and ZFP36L1 in T cells does not promote immunopathology, but rather increases resilience following infection with a pathogenic virus.

The presence of viral RNA in lung tissues of dKO mice following IAV/PR8 infection is reduced throughout the infection and by day 10 we found no detectable viral RNA in five of ten infected dKO mice (Fig. 1e). This is indicative of more efficient viral clearance when ZFP36 and ZFP36L1 are absent from T cells and may explain why the lungs of dKO mice contain reduced numbers and frequencies of CD8[+] T cells specific for IAV nucleoprotein peptide NP$_{(366-374)}$ compared to WT mice by day 10 after infection (Fig. 1f, g). We do not find differences in NP$_{(366-374)}$ specific CD8[+] T cells at earlier timepoints suggesting that recruitment of antigen-specific cells into the lung is not affected (Fig. 1f, g). We also observed an increased frequency of GranzymeB (GzmB) positive total CD8[+] T cells in the lungs of dKO mice compared to WT mice at four- and ten-days post-infection (Fig. 1h). In addition, as shown by the increased GzmB staining intensity, dKO CD8[+] T cells contain greater amounts of GzmB per cell (Fig. 1i).

Since Granzyme-B expression has been suggested to reflect the total pool of virus-specific cells which have migrated to the lungs, especially early in the response[26], the data suggest the potential for enhanced cytotoxic effector function by dKO CD8[+] T cells. To test the in vivo cytotoxicity of CD8[+] T cells we co-transferred into IAV/PR8 infected or non-infected mice a 1:1 mixture of C57BL/6 splenocytes labelled with a low amount of Carboxyfluorescein-succinimidyl-ester (CFSE) and CFSE-high splenocytes loaded with IAV NP$_{(366-374)}$ peptide. The specific killing of peptide-loaded cells is dependent upon IAV/PR8 infection and is greater in dKO mice compared to WT mice (Fig. 1j). Taken together, these data show that the absence of ZFP36 and ZFP36L1 in T cells augments the cytotoxic effector function of MHC-I restricted CTL in vivo.

**Suppression of effector cell differentiation and function mediated by ZFP36 and ZFP36L1 is cell intrinsic.** To examine the intrinsic properties of naïve dKO CD8[+] T cells following adoptive transfer we generated mice with the OT-I TCR transgene which recognises the OVA$_{257-264}$ peptide (SIINFEKL) presented by H-2K[b]. For subsequent transfer experiments to study the biology of OT-I transgenic CD8[+] T cells we used OT-I Zfp36[fl/fl]/Zfp36l1[fl/fl]CD4[wt] littermate mice as controls. Analysis of Cre recombinase expression in CD4[cre] OT-I transgenic mice showed the absence of Cre protein in CD8[+] T cells after the thymic single positive stage (Supplementary Fig. 1a, b). Following transfer of naïve OT-I cells into CD45.1[+] B6.SJL (Ptprc[a] Pepc[b/BoyJ]) mice and infection with influenza A virus/WSN/33 expressing the SIINFEKL peptide (IAV/WSN-OVA) the descendants of transferred cells are identified as CD45.2[+]. Strikingly, in mice that had received 200 naïve OT-I cells lacking both Zfp36 and Zfp36l1 the body weight loss that accompanies IAV-infection is significantly reduced compared to mice which received the same number of WT cells (Fig. 2a). Consistent with our studies in intact mice there are fewer dKO OT-I cells in the

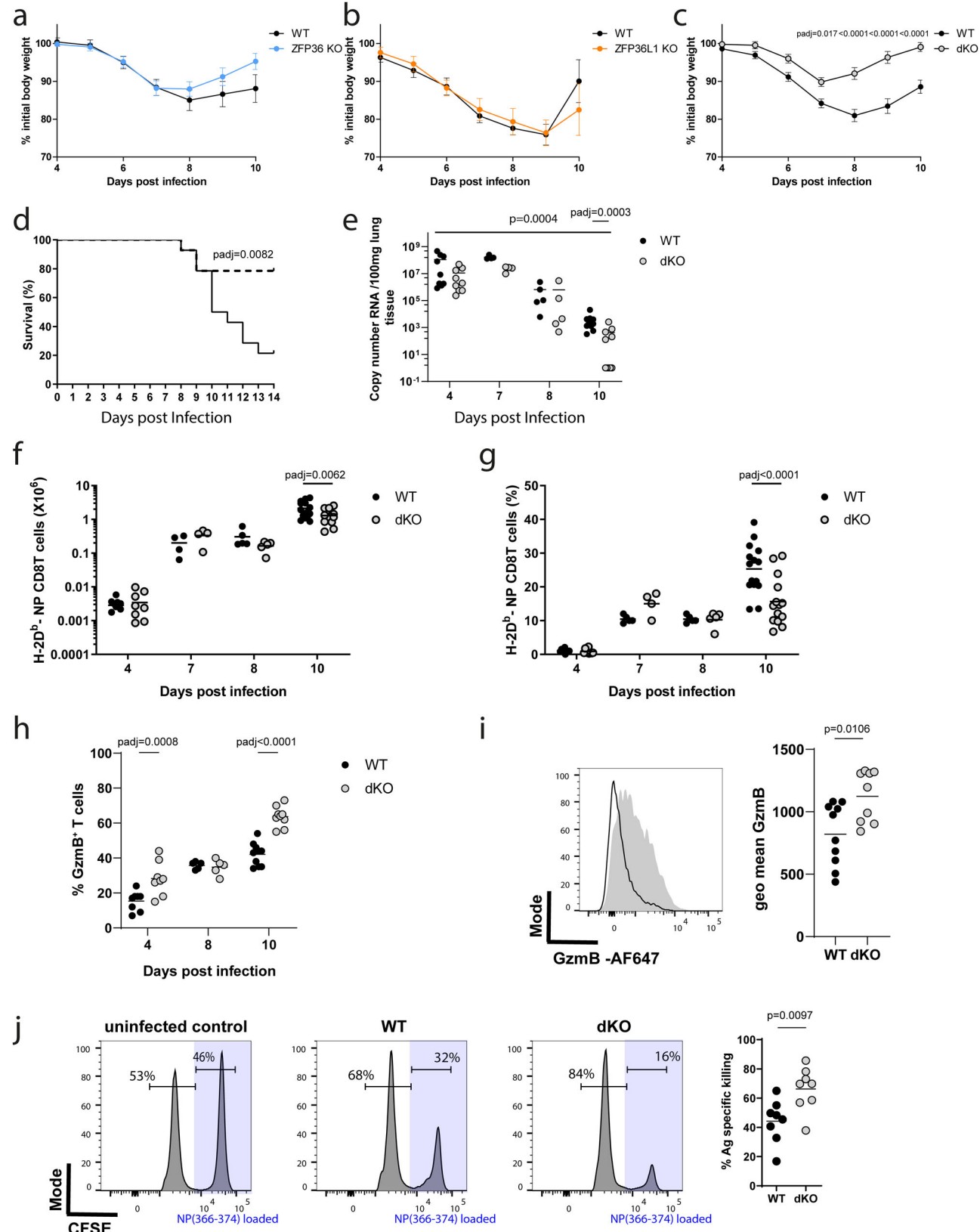

lung and the spleen at day 10 of infection than in mice with transferred WT OT-I cells (Fig. 2b). In the lungs the progeny of transferred dKO cells are skewed towards a KLRG1$^+$ IL7R$^-$ short lived effector (SLEC) phenotype compared to WT cells 10 days after infection (Fig. 2c, Supplementary Fig. 1c). At the same time frequencies of KLRG1$^-$ IL7R$^+$ memory precursor

effector cells (MPEC) were reduced in dKO CD8$^+$ T cells (Fig. 2c). In line with this, the dKO OT-I cells in the lung (Fig. 2d) and spleen (Fig. 2e, Supplementary Fig. 1d) of infected mice at day 10 contained higher frequencies of TNF- and IFNγ-producing cells than WT OT-I cells. Moreover, frequencies of Granzyme-B positive dKO cells are increased in lung (Fig. 2f,

**Fig. 1 ZFP36 and ZFP36L1 limit the anti-viral CD8$^+$ T cell effector response.** Body weight loss of initial body weight (100%) in (**a**) ZFP36 KO; (**b**) ZFP36L1 KO; or (**c**) dKO mice compared with age matched WT mice. **a**, **b** Show combined data from two independent experiments with WT $n = 10$, ZFP36KO $n = 11$ in (**a**) and WT $n = 9$, ZFP36L1KO $n = 10$ in (**b**); for (**c**) data is combined from four experiments with a total $n = 20$ per genotype. Statistical significance determined by two-way ANOVA followed by Sidak's multiple-comparisons test: Error bars indicate SEM. **d** Survival curve of WT and dKO mice infected with a lethal dose of influenza A virus; $n = 14$ per genotype combined from two independent experiments. Statistical significance was tested by Mantel Cox rank test. **e** Viral-load in the lungs of mice following sub-lethal IAV infection. $n = 9$ per genotype for day 4 and 7, $n = 5$ for day 8, and $n = 10$ per genotype for day 10; Statistical significance was determined using a two-way ANOVA on log transformed data followed by Sidak's multiple-comparisons test. Adjusted $p$ values and $p$ value for the main Genotype effect are shown. **f**, **g** Absolute and relative numbers of virus specific CD8$^+$ T cells in lungs of mice on indicated days after infection. Data is compiled from three independent experiments. **h** Frequency of GzmB positive CD8$^+$ T cells recovered from lungs on different days after infection. Statistical significance in (**f–h**) was determined by mixed-effects model followed by Sidak's multiple-comparisons test. **i** Left) Representative overlay flow cytometry plot of GzmB staining in CD8$^+$ T cells on day 10 post infection. Open histogram represents WT and filled histogram dKO cells. Right panel: summary data for GzmB staining. Statistical significance determined by a two-tailed unpaired Student's $t$-test. **j** In vivo cytotoxicity assay: Representative histograms showing proportions of labelled cells following transfer into uninfected C57BL/6 mouse (PBS) or day 10 IAV infected mice of the indicated genotype. Statistical significance determined by a two-tailed unpaired Student's $t$-test. In all panels closed circles represent the WT group and open circles the dKO. Source Data are provided as Source Data file.

Supplementary Fig. 1e) and spleen (Fig. 2g) of infected mice compared to WT. This indicates the enhanced effector differentiation of naïve CD8$^+$ T cells lacking *Zfp36 and Zfp36l1* is independent of their function in other cells.

We used an adoptive transfer model with attenuated *Listeria monocytogenes* (attLm) expressing OVA to monitor in the same mouse the kinetics of differentiation of naïve OT-I cells. On the day following transfer of naive OT-I cells into B6.SJL mice we infected them by the intravenous route and sampled cells in the blood thereafter. The dKO OT-I cells accumulate slightly fewer cells than WT OT-I (Supplementary Fig. 2a) with the population skewed towards SLEC with a reduced proportion of MPEC (Supplementary Fig. 2b, c). After peptide re-stimulation ex vivo, dKO OT-I cells show an increased proportion cells that produce TNF and IFNγ, compared to WT OT-I cells (Supplementary Fig. 2d, e). Furthermore, dKO OT-I cells produced greater amounts of TNF per cell than the WT OT-I (Supplementary Fig. 2f). Thus, in model viral and bacterial infections, *Zfp36 and Zfp36l1* function within CD8$^+$ T cells to delay the onset of differentiation and limit the magnitude of acquired effector functions.

**ZFP36 and ZFP36L1 determine the CD8 effector program early after T cell activation.** It has been previously shown that ZFP36 and ZFP36L1 are transiently expressed in naive CD8$^+$ T cells after stimulation with anti CD3 and anti CD28 antibodies[20]. We find that restimulation of in vitro differentiated CTL by antigen induces a similar transient expression of ZFP36 and ZFP36L1 with peak expression at 3-4 h after stimulation (Supplementary Fig. 3a, b). Thus, to discriminate the function of the RBP during activation of naïve CD8$^+$ T cells from their function in limiting effector function in differentiated CTL, we activated OT-I cells in vitro with peptide in the presence of IL-2 and IL-12 for 24 h and cultured them with IL-2 for seven days to generate CTL. We generated CTL from OT-I-*Zfp36*$^{fl/fl}$ /*Zfp36l1*$^{fl/fl}$*CD4*$^{cre}$ mice where ZFP36 and ZFP36L1 are absent during initial T cell activation. We compared these with CTL generated from OT-I-*Rosa26L-stop_L-Cas9GFP-CD4*$^{cre}$ mice transduced at 24 h by viruses expressing sgRNAs targeting *Zfp36* and *Zfp36l1*. In the latter, both RBPs are present in OT-I cells during initial activation but absent in CTLs (Fig. 3a). Consistent with this expectation the efficiency of knockout of ZFP36 and ZFP36L1 with CRISPR-Cas9 was very high as evident from Western blotting of CTL whole cell lysates (Supplementary Fig. 3c–f). The CTL generated from OT-I-*Zfp36*$^{fl/fl}$ /*Zfp36l1*$^{fl/fl}$*CD4*$^{cre}$ mice show increased cytotoxicity against SIINFEKL peptide-loaded EL4 cells compared to that of CTL derived from WT mice (Fig. 3b; Supplementary Fig. 4a). The CTL numbers recovered after the 3 h killing assay were the same for WT and dKO CTLs, suggesting a higher efficiency of killing on a per

cell basis by the dKO CTL (Supplementary Fig. 4b). Moreover, dKO CTLs were also more efficient in killing target cells loaded with a lower affinity peptide SIIVFEKL (V4), highlighting their increased potency when triggered by weaker signals (Supplementary Fig. 4c).

Notably, the CTL generated by Cas9-mediated knockout of *Zfp36* and *Zfp36l1* were not more potent than CTL generated following transduction with viruses expressing non-targeting sgRNAs (Fig. 3c). Upon T cell stimulation the expression of Granzyme-B and TNF is increased when *Zfp36* and *Zfp36l1* were absent before T cell activation (Fig. 3d, e). Notably, CTL produced increased TNF upon restimulation irrespective of the mode of genetic modification (Fig. 3f, g) indicating that the RBP directly target and limit TNF in differentiated CTL. By contrast, there was no difference in the amount of Granzyme-B in CTL when ZFP36 and ZFP36L1 were absent because of Cas9 mediated deletion. CTLs generated from OT-I-*Zfp36*$^{fl/fl}$ /*Zfp36l1*$^{fl/fl}$*CD4*$^{cre}$ mice show increased Granzyme-B abundance already prior to stimulation (Supplementary Fig. 4d). We did not observe differences in IFNγ production by CTL, while there is an increase in frequency of IL2$^+$ cells, but not of IL2 produced per cell, suggesting that the posttranscriptional regulation of these cytokines differs according to the differentiation status of the T cell (Supplementary Fig. 4e, f). In summary, these findings suggest that the increased cytotoxicity and Granzyme-B expression of CTL lacking *Zfp36* and *Zfp36l1* is not due to the absence of the RBP in CTL per se, it reflects rather the specific function of these RBP during early activation in naïve T cells as repressors of the CTL differentiation program.

**Dynamic gene expression networks targeted by ZFP36 and ZFP36L1 early after CD8$^+$ T cell activation.** To identify the direct targets of the ZFP36-family which act early in CD8$^+$ T cells to regulate the CTL differentiation we employed an approach that was agnostic to whether this was mediated by effects on RNA stability, localisation or translation. We searched for transcripts strongly (log2FC > 1.3 or <−1.3; $p$. adj. Value <0.05) induced or repressed following CD8$^+$ T cell activation for 6- and 18-h[27]. To identify direct targets of the RBP we performed ZFP36L1 "individual-nucleotide resolution UV crosslinking and immunoprecipitation" (iCLIP) on OT-I CTLs stimulated with peptide for three hours using a specific antibody for ZFP36L1. We combined this data with a published data set which used a pan-ZFP36 family antibody for high-throughput sequencing of RNA isolated by crosslinking immunoprecipitation (HITS-CLIP) from CD4 T cells, which were activated for 4 h[20]. This yielded a unified list of 1147 candidate genes which were present in either or both data sets. The intersection of the dynamically regulated transcripts in CD8$^+$ T cells with this list identified 204 candidates for direct

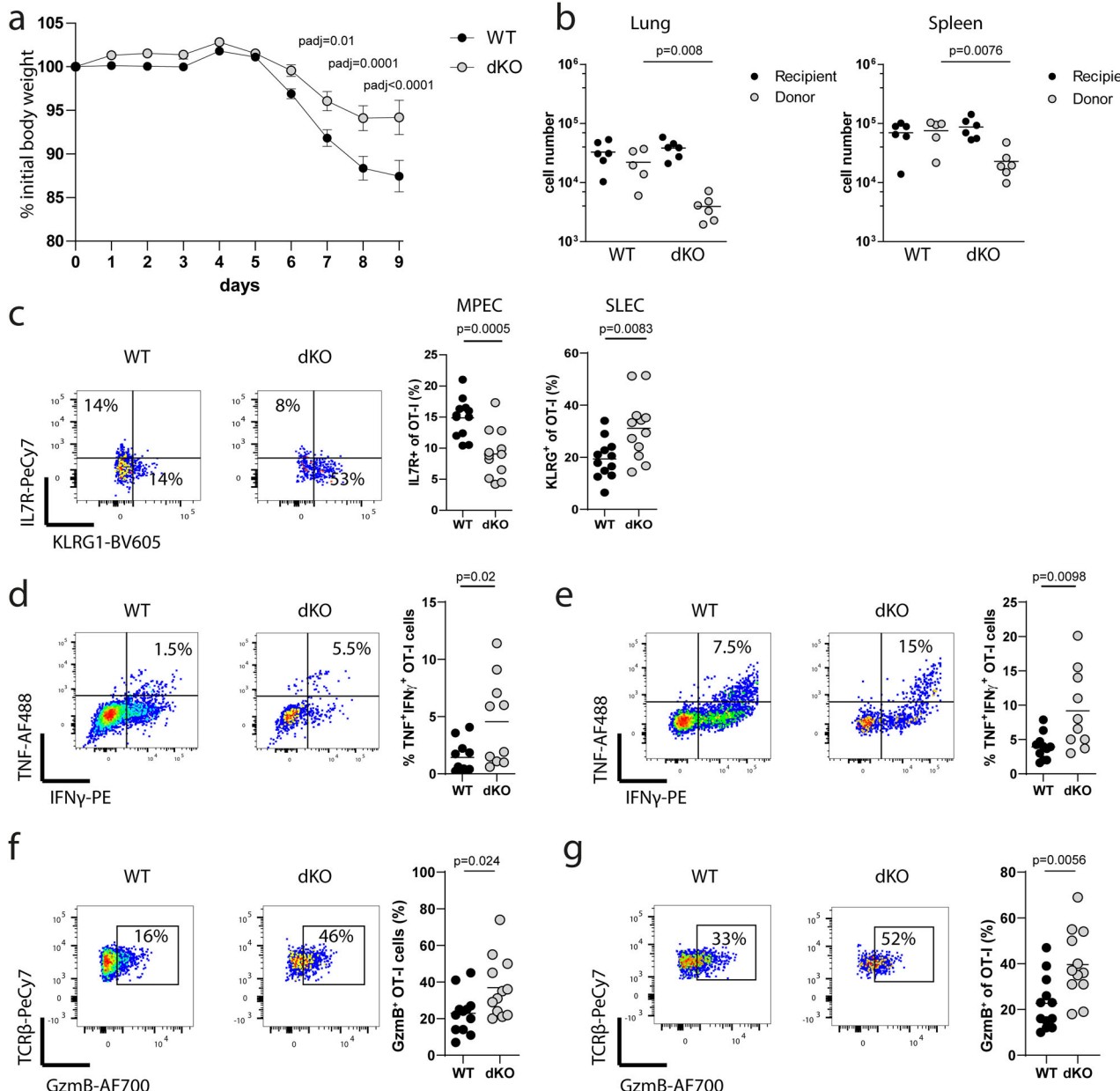

**Fig. 2 Suppression of effector cell differentiation and function mediated by ZFP36 and ZFP36L1 is cell intrinsic. a** Bodyweight loss of WT mice which received 200 WT or dKO naïve OT-I cells the day before infection with a sublethal dose of WSN-OVA virus. Data is compiled from four independent experiments with the following number of mice per day: day 0-3 WT $n = 12$, dKO $n = 11$; day 4-8 WT $n = 29$, dKO $n = 30$; day 9 WT $n = 20$, dKO $n = 23$; Statistical significance was tested using two-way ANOVA followed by Sidak's post-test for multiple comparisons; error bars indicate SEM. **b** Numbers of OVA$_{257-264}$ peptide tetramer-binding CD8$^+$ T cells in the lung and spleen of CD45.1 recipient mice following transfer of WT or dKO OT-I CD8$^+$ T cells on day 10 post IAV infection. **c** Representative flow cytometry and frequencies showing relative abundance of MPEC and SLEC among transferred OT-I cells recovered from lungs on day 10 post infection. Data is compiled from two independent experiments. Representative flow cytometry and frequencies of TNF and IFNγ producing OT-I cells in (**d**) lung and (**e**) spleen 10 days after infection. Representative flow cytometry plot and frequencies of GzmB positive cells among OT-I cells in the lung (**f**) and spleens (**g**) 10 days after infection. Data is compiled from two independent experiments. Statistical significance in panels (**b**–**g**) was tested using a two-tailed unpaired Student's t-test. In all panels closed circles are the WT group and open circles the dKO group with each symbol representing data from an individual mouse. Source Data are provided as Source Data file.

regulation by ZFP36 and ZFP36L1. Targets induced at 6 and/or 18 h post activation comprised the majority with 166 genes (81%) of which 67 transcripts are shared between both time points (Fig. 4a).

The target mRNAs could be broadly clustered into five groups according to their expression dynamics following activation (Fig. 4b; Supplementary Fig. 5a, b; Supplementary Data 1). The group I cluster includes genes that are induced early and

transiently following T cell activation and include transcription factors of the NF-κB pathway and genes involved in signalling and protein synthesis (Supplementary Fig. 5c). This group also includes genes involved in metabolism (*Slc7a1/5*) and *Kdm6b* encoding a Histone-lysine specific demethylase. Group II comprises genes induced by six hours and maintained high expression at 18 h; it contains cytokines and chemokines which include *Il2*, *Ifng*, *Tnf*, *Ccl3* and *Ccl4*. Genes involved in protein

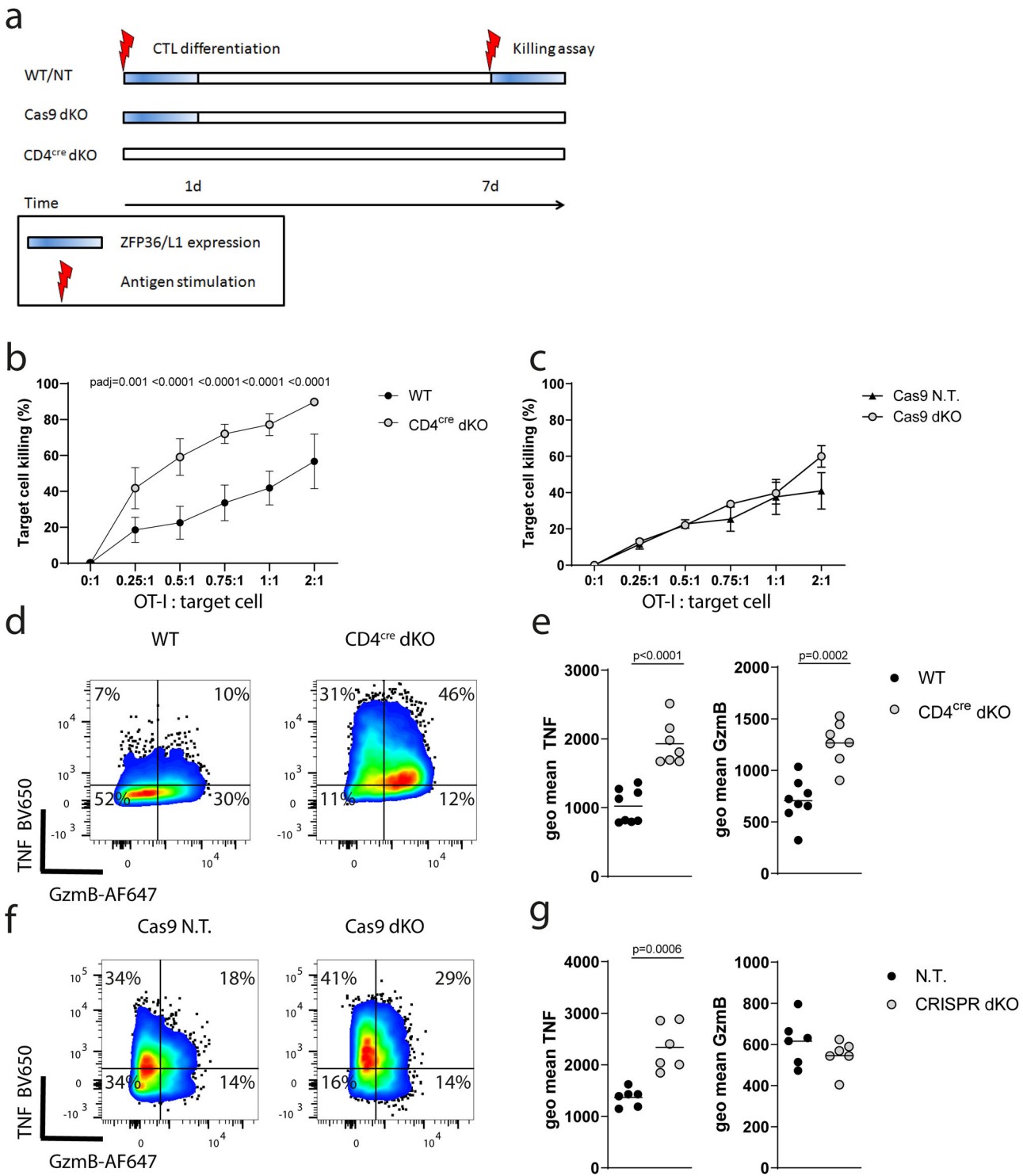

synthesis, signaling and transcription factors such as *c-Myc*, *Zeb2*, *Irf4* and *Irf8*, which are important for promoting CTL differentiation, are also represented in this group. Prominent amongst genes induced 18 h after activation in group III were ZFP36-family bound mRNAs encoding *Cdk1*, *Cdk6* and *Cdc45* which are associated with cell division. Cluster IV and V contain candidate target genes which are repressed by 6 h or 18 h post activation respectively, relative to their expression in non-activated naïve cells. Some of these targets such as *Tagln2* or *Vim* are functionally associated with signalling and cytoskeleton remodelling. The transcription factors *Fos* and *Gata3* as well as

*Zfp36* are also among group V of target mRNAs being repressed at 6 h and 18 post activation (Supplementary Data 1). *Fos* and *Zfp36* are well-known immediate early genes and present in the clusters of repressed genes because of their strong and transient transcriptional induction at timepoints earlier than six-hours. These data suggest that the RBPs can act consecutively on multiple functionally diverse target transcripts over the first 24 h of activation. As the effector differentiation is accelerated in the absence of ZFP36 and ZFP36L1 we focussed on the direct regulation of genes linked to the acquisition of effector function.

**Fig. 3 ZFP36 and ZFP36L1 determine the CD8 effector program early after T cell activation. a** Experimental design: In WT cells or cells transduced with non-targeting guides the ZFP36 and ZFP36L1 proteins are transiently expressed after T cell activation prior to transduction as well as upon restimulation of CTL. Cells which have undergone Cas9 mediated deletion of ZFP36 and ZFP36L1 lack both proteins upon restimulation but not during the initial activation period. CD4cre mediated deletion generates naïve CD8+ T cells which lack ZFP36 and ZFP36L1 during initial activation and re-stimulation. **b** In vitro killing of N4 peptide loaded EL4 cells after three hours at indicated CTL to EL4 ratios, using CTLs with CD4cre mediated deletion of ZFP36 and ZFP36L1 ($n = 6$) and WT CTLs ($n = 7$). **c** Killing assay using CTLs which were transduced with non-targeting guides ($n = 3$) or guides targeting both ZFP36 and ZFP36L1 ($n = 3$) is shown. For (**b**) and (**c**) data is represented as mean values ± SD. Statistical significance was determined by one-way ANOVA followed by Tukey's test for multiple comparisons. **d** Representative flow cytometry plots showing GzmB and TNF expression in CTLs after three hours of WT and CD4cre dKO CTL exposure to peptide loaded EL4 cells at a 1:1 ratio. **e** Plots showing geometric mean fluorescence intensity for GzmB and TNF as in (**d**), gated on the positive population. Each dot represents a biological replicate from four independent experiments. **f** Representative flow cytometry plots showing GzmB and TNF expression in CTLs treated with non-targeting and ZFP36 and ZFP36L1 specific guides, after three hours of exposure to peptide loaded EL4 cells at a 1:1 ratio. **g** Plots showing geometric mean fluorescence intensity for GzmB and TNF as in (**f**), gated on the positive population. For sgRNAs targeting ZFP36 and ZFP36L1 or non-targeting controls the data is representative of two independent experiments, each with three independent pairs of sgRNAs. Statistical significance was determined by a two-tailed unpaired Students t-test. In all panels filled circles represent the respective WT control and open shaded circles the dKO. Source Data are provided as Source Data file.

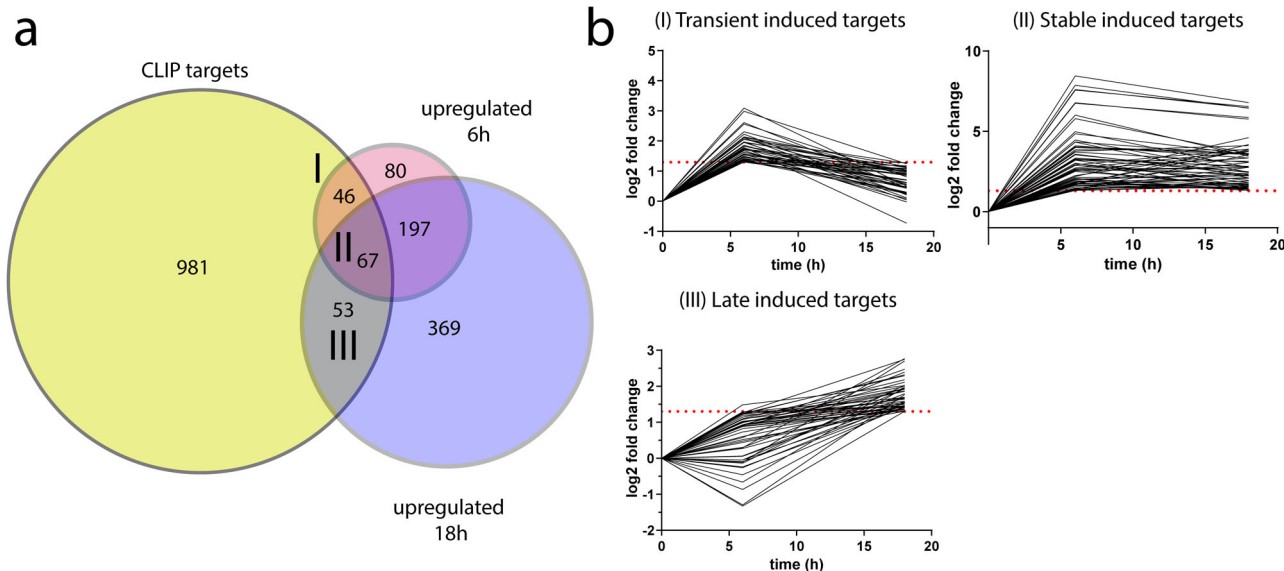

**Fig. 4 Dynamic gene expression networks targeted by ZFP36 and ZFP36L1 early after CD8+ T cell activation. a** Overlap of transcripts significantly differentially expressed (adjusted p value <= 0.05) by log2 FC > 1.3 or < −1.3, following activation of naive CD8+ T cells for 6 or 18 h (analysed by DESeq2; Wald test followed by Benjamini–Hochberg correction for adjusted p values) with CLIP targets of ZFP36 and ZFP36L1. **b** Clusters I–III showing ZFP36 family mRNA target dynamics during T-cell activation. Dotted line shows log2 FC = 1.3. Source Data are provided as Source Data file.

**ZFP36 and ZFP36L1 limit the magnitude of IL-2 production.** IL-2, which is an important cytokine for effector T cell differentiation, was identified as a direct target of ZFP36 and ZFP36L1 in the CLIP analysis (Supplementary Fig. 6a). However, the regulation of cytokines by ZFP36 has been shown to be cell context dependent[23] and previous studies found that the absence of *Zfp36* in CD8+ T cells did not affect IL-2 production in CD4+ or CD8+ T cells[20,24]. In contrast, another study reported enhanced *Il2* mRNA stability in mixed splenocyte and T cell populations from Zfp36−/− mice[28]. Therefore we sought to examine IL-2 production by dKO cells using a sensitive bispecific antibody-based capture method[29] which measured IL-2 secretion at the single cell level in the absence of a secretion inhibitor. Following peptide-stimulation the frequency of WT and dKO CD8+ T cells producing IL-2 peaked around six-hours following peptide-mediated T cell activation. (Fig. 5a, b). We found a substantial increase in the proportion dKO OT-I cells producing IL-2 compared to WT OT-I cells during the first 16 h (Fig. 5a, b). However, by 24 h the frequency of IL-2 expressing cells is low and does not differ between dKO and WT cells. Therefore, the suppressive mechanisms that lead to the cessation of IL-2 production are intact in the absence of the RBP. In contrast we find the same staining intensity for CD25

between WT and dKO OT-I cells during the first 16 h after activation. However, expression of CD25 persists on dKO cells at 24 h a time at which it shows diminished expression on WT cells (Fig. 5c). For the WT cells the intensity of IL-2 staining showed little difference at any of the timepoints tested before 24 h after activation. This is indicative of the all-or-none digital response to T cell stimulation. By contrast, at three and six hours after activation, OT-I cells lacking ZFP36 and ZFP36L1 stain more intensely than WT OT-I for IL-2 indicating that they produce more IL-2 per cell (Fig. 5d). Thus, early after activation ZFP36 and ZFP36L1 limit the frequency of activated T cells producing IL-2 and the amount of IL-2 produced by the activated CD8+ T cell.

To investigate in detail how the RBPs regulate IL-2 expression we measured the *Il2* mRNA and found increased amounts in dKO CD8+ T cells, especially three- and six-hours after stimulation with peptide (Fig. 5e). To assess whether the increased abundance in *Il2* mRNA is due to increased stability in the absence of the RBPs we activated naive OT-I WT and dKO cells for 3 h and treated them with Triptolide a global transcription inhibitor. The inhibition of transcription allowed us to assess the decay of *Il2* (Fig. 5f). The absence of both RBPs increased the stability of *Il2* mRNA. To understand whether

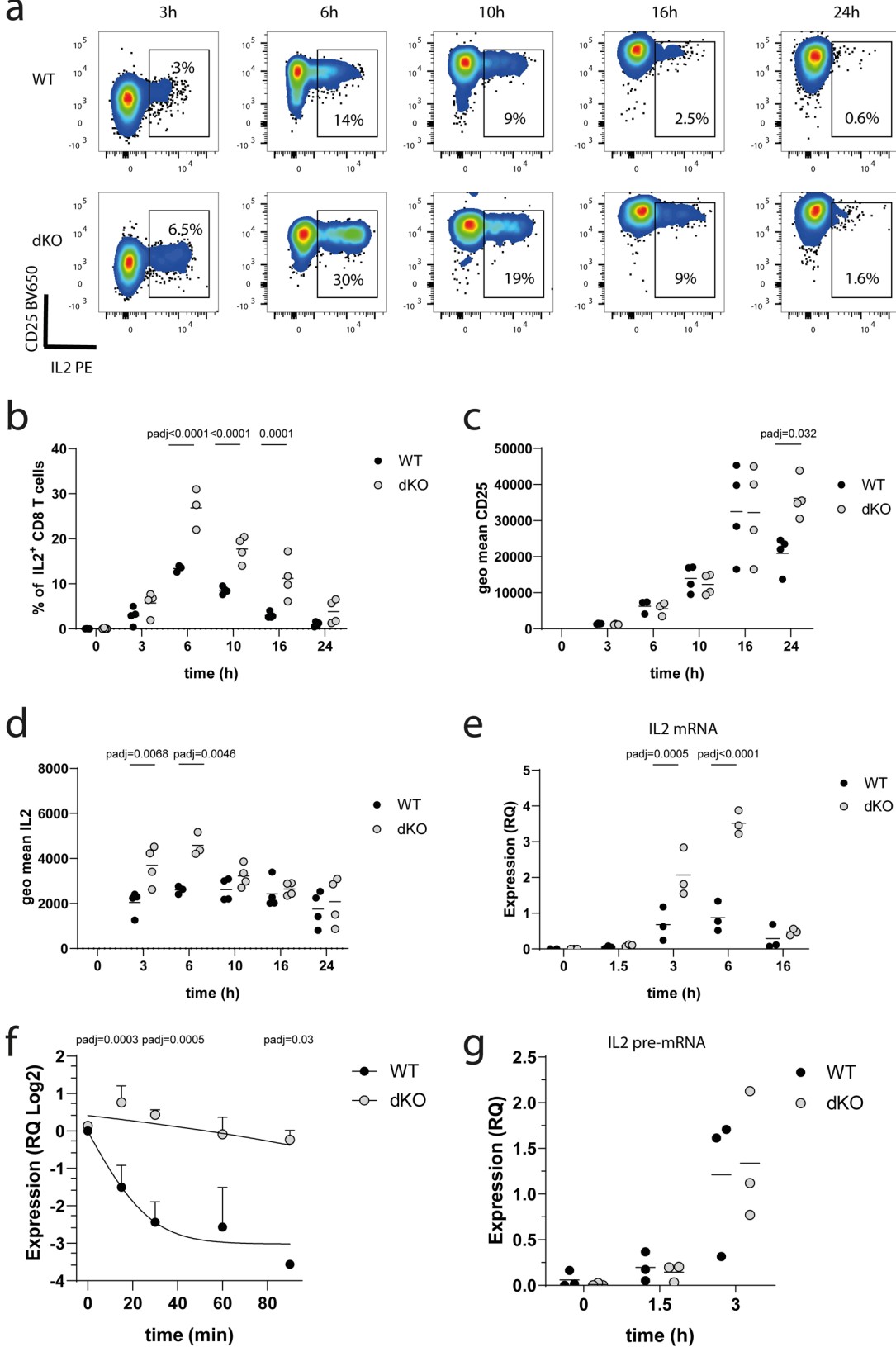

absence of the RBPs also alters the transcription of the *Il2* gene, we measured the unprocessed *Il2* pre-mRNA. Although this is not a direct measure of transcription, mRNA transcripts with retained introns are recently transcribed. However, we did not find differences in pre-mRNA transcripts between WT and dKO naive CD8[+] T cells early after activation (1.5 and 3 h). Together

these results suggest that *Il2* mRNA is regulated directly by ZFP36 and ZFP36L1 (Fig. 5g).

**Increased autocrine IL-2 production is not necessary for differentiation into SLEC.** The regulation of cytokines by ZFP36

**Fig. 5 ZFP36 and ZFP36L1 suppress IL-2 production. a** Representative flow cytometry plots showing IL-2 and CD25 staining intensity by naïve OT-I cells stimulated with N4 peptide for the indicated times. **b** A plot showing frequency of IL-2 producing OT-I cells over time; (**c**) Geometric mean of CD25 fluorescence per cell over time. **d** Geometric mean of IL-2 fluorescence per cell over time. Graphs in (**a–d**), are compiled from three independent experiments. Each point represents a biological replicate. **e** Relative quantity of IL2 mRNA expression measured by real time PCR, in naïve WT and dKO OT-I cells which were stimulated for the indicated times with N4 peptide. Relative quantities of IL2 mRNA to RPL32 mRNA are shown. **f** Relative quantity of IL2 mRNA expression, measured by real time PCR, in naïve WT and dKO OT-I cells stimulated for 3 h with N4 peptide and treated with Triptolide for indicated times. A linear regression model of one phase decay was fitted on the data. Data is presented as mean values ± SD from $n = 3$ biological replicates. Statistical significance was determined on non-transformed data. **g** Relative quantity of unprocessed IL2 pre-mRNA expression in naïve WT and dKO OT-I cells stimulated with N4 peptide for indicated times. IL2 pre-mRNA expression was normalized to RPL32 mRNA expression. Each point represents a biological replicate. Statistical significance in (**b–g**) was determined using a mixed-effects model analysis followed by Sidak's post-test for multiple comparisons. In all panels filled circles represent the respective WT control and open shaded circles dKO. Source Data are provided as Source Data file.

and ZFP36L1 can influence other cells of the immune system and result in an altered inflammatory milieu which influences effector cell differentiation[12,13,16]. Indeed, IL-2 has been proposed to promote effector differentiation[30]. To test whether the increased production of IL-2 promotes rapid formation of SLEC, we co transferred 500 naïve WT and 500 dKO OT-I cells, electroporated with a complex of Cas9 protein and a non-targeting guide RNA, into CD45.1 recipients and infected them with att*Lm-OVA* the following day. In another group of recipients, we co-transferred 500 naïve WT OT-I cells together with 500 dKO OT-I, electro-porated with Cas9 protein and a guide RNA targeting the *Il2* gene. The dKO OT-I cells showed higher frequencies of KLRG1+ SLEC in blood of the recipients on day 5 post infection irrespective of their ability to produce autocrine IL-2 (Supplementary Fig. 6 b–e). Thus, we conclude that dKO cells which are in the same inflammatory environment as WT cells still differentiate more into SLEC in the absence of autocrine IL-2.

**ZFP36 and ZFP36L1 directly limit the expression of NF-κB, IRF8 and Notch1.** Multiple transcripts encoding components of the canonical and noncanonical NF-κB pathway, including *Nfkb1, Nfkb2, Nfkbib* and *Rel* are induced at 6 h post-activation and cluster in group I. The NF-κB pathway is critical to integrate TCR signaling with CD28 costimulation and inflammatory cues. In concert with the NFAT/AP1 complex, NF-κB promotes the induction of IL-2 by T cells[3]. Analysis of ZFP36-family interactions with RNA confirmed *Nfkb1* (Supplementary Fig. 6f), *Nfkb2* (Fig. 6a) and *Rel* (Fig. 6b) mRNA to be directly bound with accumulation of reads at AU-rich elements (AREs) in their 3′ UTRs that are highly conserved between mammalian species (Fig. 6c, d: Supplementary Fig. 6f). Consistent with this interaction being consequential, we found that in CD8+ T cells from dKO mice the abundance of representative members of the non-canonical and canonical NF-κB pathways, NF-κB2 (p100) (Fig. 6e) and cREL (Fig. 6f) is greater compared to WT following activation with anti-CD3. While NF-κB2 (p100) abundance remained greater in the dKO than the WT over the whole course of T cell activation, increased cREL protein was observed in the dKO only at early time-points.

The NF-κB pathway has been shown to interact with other transcription factors including NOTCH1 to drive differentiation into CTL by directly regulating the expression of the transcription factor Eomesodermin (EOMES)[31] and Granzyme-B[32]. The *Notch1* mRNA is bound by ZFP36 and ZFP36L1 according to T cell CLIP data (Supplementary Fig. 7a). This is consistent with an earlier biochemical demonstration of the interaction of ZFP36L1 with a 61 nucleotide sequence containing the *Notch1* AREs[33]. Although *Notch1* is not among the mRNAs in the induced clusters, the expression of the NOTCH1 intracellular domain is increased in naïve dKO CD8+ T cells at three-, six- and

18-hours post stimulation compared to WT cells (Supplementary Fig. 7b, c).

TCR induced EOMES expression is NF-κB and NOTCH-1 dependent and mediates the effector and memory differentiation program in CD8+ T cells by promoting cytokine and granzyme expression[31,32,34,35]. We found dKO naïve CD8+ T cells activated with plate bound anti-CD3 antibody showed greatly increased frequencies of Granzyme-B and EOMES positive cells compared to WT cells 72 h after activation (Fig. 6g, h). The increased expression of EOMES three days after activation was not due to relief of direct inhibitory effects of the ZFP36 -family on the *Eomes* mRNA as it was not identified as a target mRNA. EOMES becomes highly expressed during the later course of the immune response and not in naïve T cells[31]. This data highlights the more rapid acquisition of an effector state in dKO cells mediated via targets accelerating EOMES expression. Group II transcription factors *Irf4* and *Irf8* have also been shown to be important for CTL differentiation and function[36,37]. In particular *Irf8* has been suggested to act independently of T-box transcription factors *Tbet* and *Eomes* to promote effector functions[36]. *Irf8* mRNA, which is rapidly induced upon T cell activation and continues to be highly expressed by 18 h, is also bound by ZFP36 and ZFP36L1 in T cells (Supplementary Fig. 7d). IRF8 protein is increased in dKO cells compared to WT cells, particularly so at later time points of stimulation (Supplementary Fig. 7e, f). Taken together, these data suggest a collective of transcription factors known to form a network promoting T cell differentiation are regulated by ZFP36 and ZFP36L1.

**ZFP36 and ZFP36L1 inhibit TCR mediated T cell activation and promote dependence on CD28.** T cell activation, proliferation and differentiation are intimately linked. A previous study suggested CD8+ T cells deficient for ZFP36 to proliferate more rapidly in response to TCR stimulation[20]. Moreover, our CTL killing assays with low affinity peptide suggested a reduced activation threshold for dKO CTL. To test whether naïve isolated CD8+ T cells were more sensitive to TCR dependent activation we titrated the amount of plate-bound anti-CD3 and analyzed the proliferation of cell trace labeled dKO and WT cells after 72 h. We found that dKO cells responded to lower doses of anti-CD3 with greater proliferation as compared to WT cells. Interestingly, upon stimulation with higher doses of anti-CD3 the dKO cells had still a proliferative advantage as compared to their WT counterparts (Fig. 7a–c).

Costimulation by cytokines and cell surface ligand-receptor-interactions e.g., CD28 is critical for naïve CD8+ T cell activation and subsequent differentiation. The dependency of T cells on CD28 is regulated by the NF-κB pathway[3] including NF-κB2[38]. Also not only transcription of *Il2* is dependent on CD28 costimulation via NF-κB[39] but CD28 also promotes the stabilization of the *Il2* mRNA[40]. This prompted us to test the

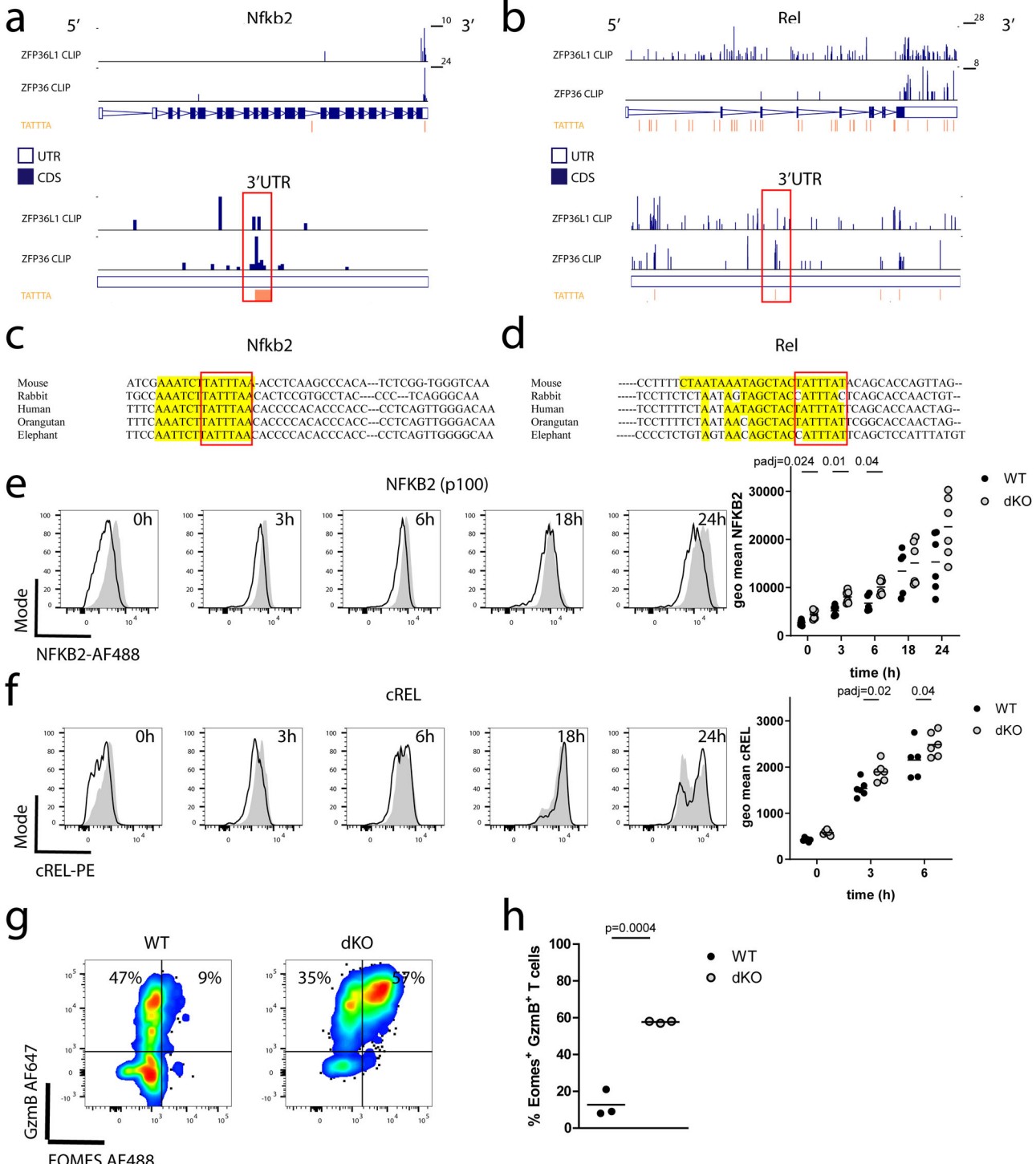

co-stimulation dependence of dKO CD8$^+$ T cells by the titration of anti-CD28 antibodies. Following stimulation with 5 μg/ml plate bound anti-CD3, cell trace labelled dKO CD8$^+$ T cells divide more than their WT counterparts (Fig. 7d). The addition of anti-CD28 (clone 37.51) increases the numbers of both, WT and dKO cells present after 72 h culture, compared to cultures with anti-CD3 alone (Fig. 7d, e). WT cells show a greater sensitivity to lower amounts of CD28 costimulation compared to the dKO CD8$^+$ T cells. The latter are only mildly responsive to the inclusion of anti-CD28 as measured by the absolute cell numbers in the cultures after 72 h with LogEC$_{50}$ of 2.2 μg/ml for WT and 1.2 μg/ml dKO cells (Fig. 7e). dKO cells accumulate

more cells per generation, when stimulated with CD3 alone as compared to their WT counterparts (Fig. 7f, g), which results in a reduced sensitivity to additional CD28 costimulation in dKO cells. This contrasts with WT cells which are critically dependent on signals from CD28, which opposes the inhibition of activation by ZFP36 and ZFP36L1, to enable a robust proliferative response.

As shown above stimulation of CD8$^+$ T cells with anti-CD3 alone induced significantly greater amounts of both NF-κB2(p100) (Fig. 7h, i) and IRF8 (Fig. 7j, k) in dKO cells compared to WT. The addition of 5 μg/ml plate bound anti-CD28 augmented the expression of both transcription factors, indicating they were responsive to co-stimulation and reduced the

**Fig. 6 ZFP36 and ZFP36L1 limit the early expression of transcription factors driving effector differentiation. a** CLIP data showing sequencing reads across (**a**) *Nfkb2* and (**b**) *Rel* transcripts (a set of top two lanes) with an expanded view of the 3′UTR (a set of bottom two lanes). In each set top lane shows ZFP36L1 CLIP data from OT-I CD8 CTLs stimulated for 3 h with N4 peptide; bottom lane shows ZFP36 CLIP data from in vitro activated naive CD4 T cells. TATTTA motifs are identified in orange; (**c, d**) conservation among vertebrates of the ZFP36/ZFP36L1 binding site determined using MULTIZ alignment tool in UCSC genome browser for (**c**) *Nfkb2* and (**d**) *Rel*. Binding as identified by CLIP is highlighted in yellow. Selected species are ordered from top to bottom according the evolutionary proximity of the clades. Red bracket highlights the ARE. For *Rel* the best conserved binding site is displayed. **e** Differential protein expression of NF-κB2 and (**f**) cREL by naive CD8+ T cells following activation with plate bound anti-CD3 antibody. Representative flow cytometry histograms (left panel) and geometric mean fluorescence (right panel) are shown. Open histograms represent WT and filled histograms show dKO cells. Statistical significance was determined using a mixed-effects model analysis followed by Sidak's test for multiple comparisons. Data is compiled from two of three independent experiments, with each dot representing a biological replicate. **g** Flow cytometry plots showing expression of EOMES and GzmB by naive WT and dKO CD8+ T cells activated with plate bound CD3 for 72 h. **h** Frequency of GzmB and EOMES expressing cells. Statistical significance was tested with a two-tailed unpaired Student's *t*-test. Data is compiled from two independent experiments and three biological replicates. In all panels filled circles represent the respective WT control and open shaded circles dKO. Source Data are provided as Source Data file.

differences in NF-κB2 (Fig. 7h,i) and IRF8 (Fig. 7j,k) expression between dKO cells and WT. Importantly, the increased expression of NF-κB2 and IRF8 by CD8+ T cells lacking ZFP36 and ZFP36L1 persisted upon addition of large amounts of recombinant IL-2 indicating these differences are not secondary to increased IL-2 production by the dKO T cells (Fig. 7h-k). Thus, ZFP36 and ZFP36L1 regulate CD28 co-stimulation also independently of the production of IL-2. The absence of ZFP36-family members in the naïve CD8+ T cell reduces the requirement for CD28-mediated costimulation in part by enabling the expression of costimulation-dependent transcription factors.

## Discussion

Here we show that ZFP36 and ZFP36L1 act in CD8+ T cells to reduce the tempo of CTL differentiation and to limit the potency of the resulting CTL. These properties are endowed by the absence of the RBP during thymic selection or, shortly after T cell activation by antigen when the RBP are transiently expressed. Our observations are consistent with the work of others[4,7–10] indicating that the programming of CTL fate happens early after activation and indicate that RBP are critical in limiting this process.

The enhancement of the CTL response underpins why mice in which T cells lack these RBP are more resilient to infection by IAV and why transfer of naive dKO CD8+ T cells into WT mice is sufficient to reduce the body weight loss triggered by IAV. The finding of slightly fewer, yet more potent, antigen-specific dKO effector cells ten-days after infection likely reflects accelerated viral clearance and an earlier contraction of the CTL pool. The removal of RBP that limit the production of pro-inflammatory cytokines and cytotoxic factors might have been predicted to cause increased immunopathology following IAV infection, however this was not the case. A reduced number of inflammatory CD8+ T cells may contribute to the reduced immunopathology/ body weight loss observed when T cells lack ZFP36 and ZFP36L1. However, this is clearly an oversimplification because multiple parameters link T cell numbers, inflammation, body weight loss and viral clearance; these display a non-linear relationship during influenza infection[41]. Thus, we cannot formally distinguish whether the physiologically beneficial outcome, accompanied by enhanced viral clearance, can be attributed to a more rapid differentiation of a CD8 effector population or a more potent cytotoxic function on a per cell basis. It is possible that both processes are involved. Finally, while our data suggests that ZFP36 and ZFP36L1 are redundant in limiting T cell dependent resilience to influenza, different outcomes for the individual RBP might be found upon other immune challenges such as persistent infection.

At the cellular level we identified ZFP36 and ZFP3L1 to be essential for limiting T cell activation by enforcing the dependency on CD28 co-stimulation. The regulation of the *Il2* gene by CD28 is a paradigm of TCR costimulation-regulated gene expression: CD28 promotes the stability of *Il2* mRNA, a finding over 30 years old[40] for which the mechanisms remain unresolved; CD28 also regulates gene expression via the NF-κB transcription factors which act in concert with AP-1 and NFAT to promote the transcription of the *Il2* gene. Here we show, using naïve mouse CD8+ T cells, that ZFP36 and ZFP36L1 regulate *Il2* directly via binding to IL-2 mRNA and promoting its instability. It remains possible, but unexamined here, that the RBP also limit the rate of translation of *Il2* mRNA. We speculated whether there is an early indirect transcriptional regulation of *Il2* which would be partly accounted for by the RBP limiting expression of *Nfkb2* and *Rel* subunits of NF-κB[38,42,43]. However, we find that there is no increased transcription in the absence of the RBPs. It is worth noting that the NF-kB pathway integrates multiple costimulatory signals and acts on the transcription of different genes which are important to mediate the effects of costimulation in T cells independent of its regulation of IL-2[44]. This is consistent with our results where we find that enhanced differentiation of dKO OT-I cells into KLRG1+ effector cells is likely to be independent of autocrine IL-2 production, suggesting that differentiation is regulated by the RBPs via alternative factors. Moreover, we show that during naïve CD8+ T cell activation the abundance of IRF8 and NF-κB2 is sensitive to costimulation via CD28 independently of IL-2, suggesting that the reduced requirement for costimulation and accelerated differentiation of RBP deficient CD8+ T cells are linked. The limiting activity of RBPs on cytokines and transcription factors early during activation of T cells might be crucial to ensure that a T cell response is only launched when antigen is presented by a sufficiently licenced and matured APC. In line with this, while naïve dKO CD8+ T cells show a reduced activation threshold and are less dependent on stimulation, it remains to be further investigated whether the phenotypically naïve T cells used in this study, have respondend to some extent to self-antigen during thymic selection or in the periphery, which might sensitize them for a more rapid differentiation into potent effector cells. Our current knowledge and unpublished data however indicate that the dKO naïve cells do not express effector molecules or transcription factors related to effector differentiation prior to stimulation.

While our analytical approach focussed on target genes regulated dynamically at the level of RNA at six- and 18-hours following activation will not detect targets that are dynamically regulated earlier than this, it has shown additional mechanistic insight into the pathways and processes regulated by these RBP. These include the transcription factors *Notch1* and *Irf8* that are necessary for CD8 effector formation[32,36]. *Notch1* has been shown to cooperate with the NF-κB transcription factors to induce the expression of EOMES and Granzyme- B[31,32]. These

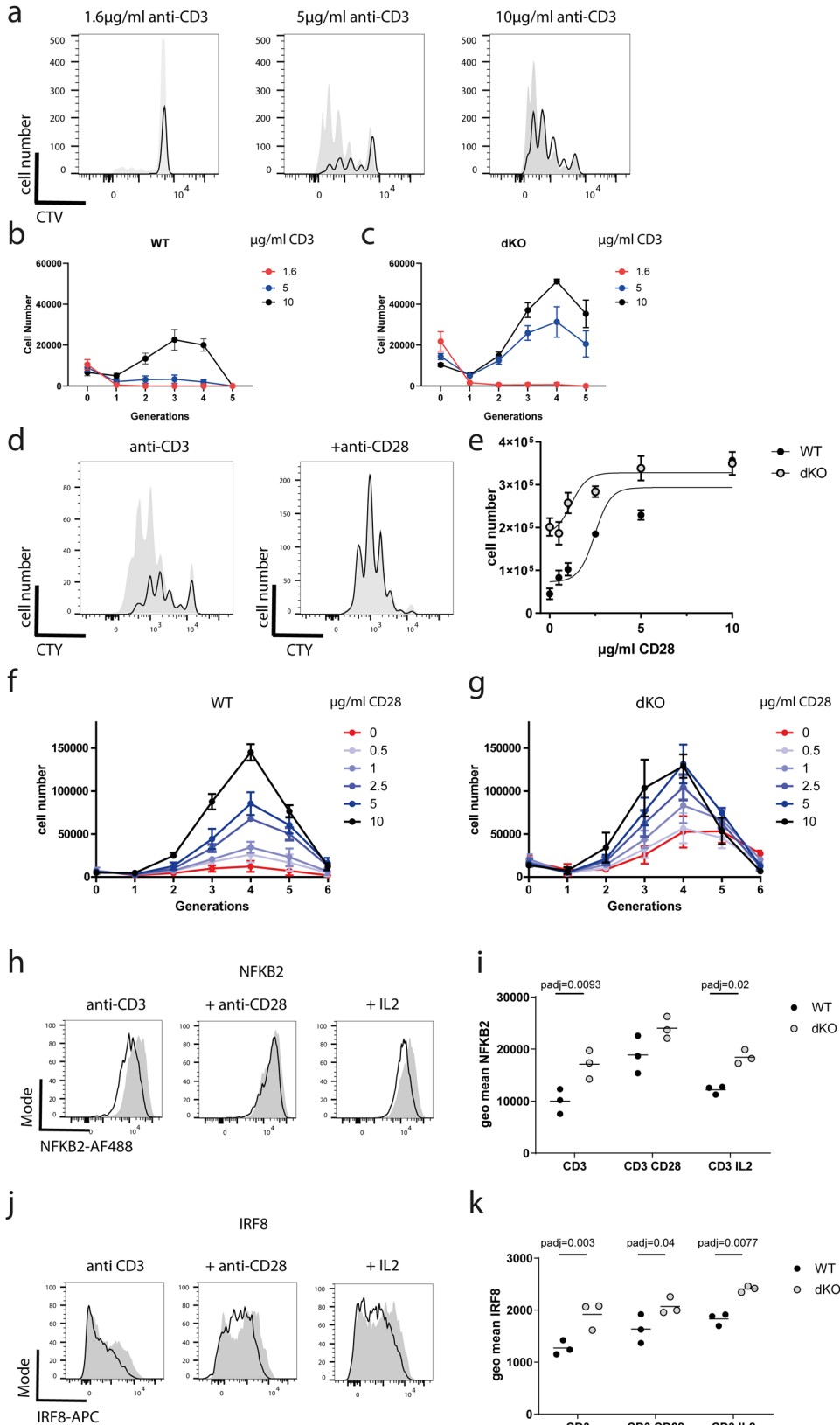

are key determinants of effector cell function and differentiation which we also find increased in the absence of ZFP36 and ZFP36L. IRF8 has been shown to promote effector differentiation by integrating antigenic and cytokine signals independently of EOMES and TBET[36]. These findings highlight the systematic nature of repression of factors driving effector differentiation by

the ZFP36 family members (Supplementary Fig. 8). We observed that protein expression of target genes *Irf8* and *Nfkb2* was more protracted in the dKO. Other targets genes- *Rel* and *Il2* show an elevated expression of protein early but the increase per cell is only transient. These differences may reflect different modes of regulation of mRNA by the RBPs (for example mRNA stability

**Fig. 7 ZFP36 and ZFP36L1 promote CD28 costimulation dependence for activation. a** CTV dilution by WT (open histograms) and dKO (filled histograms) T cells stimulated for 72 h, with indicated concentrations of anti CD3 plate bound antibody. **b**, **c** Total numbers of cells recovered per generation after 72 h of stimulation with anti CD3 antibodies. Data from $n = 3$ biological replicates is presented as mean values ± SD. **d** CTV dilution by WT and dKO T cells after 72 h in the presence of 5 μg/ml plate bound anti CD3 (left panel) and with the addition of 10 μg/ml plate bound anti CD28 (right panel). **e** Total numbers of cells recovered with varying amounts of CD28 plate bound added to 5 μg/ml plate bound CD3. Error bars indicate the SEM from three biological replicates. Non-linear regression model was fitted on the data. **f** Cell numbers per generation after 72 h with plate bound anti CD3 and varying amounts of plate bound anti-CD28 from WT and (**g**) dKO T cells. Data from $n = 3$ biological replicates is presented as mean values ± SD. **h**, **i** Representative histograms and geometric mean fluorescence showing expression of NF-κB2 and (**j**, **k**) IRF8 in naive CD8+ T cells after 24 h stimulation with plate bound anti-CD3 in the presence or absence of soluble recombinant IL-2 or plate bound anti-CD28. Open histograms represent WT and filled histograms show dKO cells. Each symbol represents a biological replicate. Statistical significance was determined with two-way ANOVA with Sidak's correction for multiple testing. In all panels filled circles represent the respective WT control and open shaded circles dKO. Source Data are provided as Source Data file.

and translational repression), but may also depend on different protein half-lives. Yet, the premature expression of one set of targets and the protracted expression of another set of target genes can be directly linked via the RBPs and result in an accelerated induction of differentiation and an earlier acquisition of effector molecules. This may as well be accompanied by the deregulation of target genes involved in anabolic metabolism and cell cycle processes as well as the restructuring of the chromatin landscape in the absence of RBP. We have not examined the latter experimentally, but we propose that ZFP36 and ZFP36L1 exert regulation across multiple cellular processes to limit T cell activation and differentiation. In this way, the RBP can coordinate the amounts of genes expressed within pathways and also integrate the activity of different pathways to regulate differentiation.

## Methods

**Mice**. Mice with single or combined floxed *Zfp36* and *Zfp36l1* alleles[33,45], Rosa26[lsl] Cas9-GFP[46], B6.Cg-Tg(CD4-cre)1Cwi mice[47] and OT-I TCR[48] (Vα2 and Vβ5 recognizing peptide residues 257-264 of chicken ovalbumin in the context of H2K^b) were generated on the C57BL/6 background at the Babraham Institute. The B6.SJL-*Ptprc^aPepc^b*/BoyJ (CD45.1) mice were bred at the Babraham Institute. CD45.1 CD45.2 double positive mice were bred as F1 from C57BL/6 (CD45.2) and B6.SJL-*Ptprc^aPepc^b*/BoyJ (CD45.1) mice at the Babraham Institute. In all experiments 8-12 weeks old male and female mice were used.

All mouse experimentation was approved by the Babraham Institute Animal Welfare and Ethical Review Body. Animal husbandry and experimentation complied with existing European Union and United Kingdom Home Office legislation and local standards. Mice were bred and maintained in the Babraham Institute Biological Support Unit. Since the opening of this barrier facility (2009), no primary pathogens or additional agents listed in the FELASA recommendations have been confirmed during health monitoring surveys of the stock holding rooms. Ambient temperature was ~19–21 °C and relative humidity 52%. Lighting was provided on a 12-h light: 12-h dark cycle including 15 min 'dawn' and 'dusk' periods of subdued lighting. After weaning, mice were transferred to individually ventilated cages with 1–5 mice per cage. Mice were fed CRM (P) VP diet (Special Diet Services) ad libitum and received seeds (e.g. sunflower, millet) at the time of cage-cleaning as part of their environmental enrichment.

**Infection with influenza virus**. 8-12-week-old female mice were used for all experiments. During infection animals were anesthetized and infected by intranasal administration with 10–50 PFU of A/Puerto Rico/8/34 (PR8 H1N1) or with 10³ PFU of OVA(257-264)- expressing influenza A/WSN/33 (WSN-OVA). Body weights were recorded daily after infection by biological support unit staff. Transfer experiments were performed by staff of the animal support unit in a blinded fashion.

Viral Load determination: RNA was extracted from infected lungs using a Ribopure kit (Ambion, USA) and cDNA was synthesized using MP specific primer: 5′-TCTAACCGAGGTCGAAACGTA -3′. The Real time PCR assay was performed with Platinum Quantitative PCR Super Mix UGD (Invitrogen, USA) by using the following primers and probe specific for Matrix protein (MP) encoding gene of influenza PR8 H1N1 virus strain. Sense primer sequence: 5′-AAGACCAATCCTGTCACCTCTGA -3′, antisense primer sequence: 5′- CAAAGCGTCTACGGCTGCAGTC -3′, and probe sequence: FAM-5′- TTTGTGTTCACGCTCACCGT-3′ TAMRA. Viral loads of the experimental samples were calculated using a standard curve made from a stock PR8 virus with a known concentration of virus plaque forming units per milliliter (PFU/ml).

Adoptive transfer experiments: Splenocytes were incubated for 30 min at 4 °C with FITC-conjugated anti-CD4, anti-CD19, anti-NK1.1, anti-F4/80, anti-CD11b, anti-CD11c, anti-MHCII Abs. Next the cells were washed and incubated for 20 min at 4 °C with 100 μl anti-FITC MACS magnetic beads (Miltenyi Biotec) per 10⁸ cells.

Cells were washed again and applied to a MACS LS column (Miltenyi Biotec) and the flow through was collected. The resulting cells were >85% CD8+ T cells as assessed by flow-cytometry. 200 naïve CD45.2+ *Zfp36^fl/fl^Zfp36l1^fl/fl^ CD4^cre^* OT-I CD8+ T cells or CD45.2+ *Zfp36^fl/fl^Zfp36l1^fl/fl^* OT-I CD8+ T cells were transferred i.v. into CD45.1+ recipient mice. One day later the recipient mice were infected with WSN-OVA.

Pulmonary and splenic single cell preparation: Mice were euthanized and lungs were perfused by injecting 5 ml of HBSS + 1 mM EDTA and 25 mM HEPES through the right ventricule of the heart. Single cell suspensions from lungs were prepared with collagenase A (Sigma-Aldrich, USA) 0.5 mg/ml and DNAse I (Sigma- Aldrich, USA) 0.15KU/ml in sterile HBSS with 5% FBS for 1 hour at 37 °C. The lung digest was filtered and mononuclear cells were re-suspended in 1% BSA and 1 mM EDTA containing PBS (FACS Buffer). Viable cell counts were determined using trypan blue exclusion. Cells were first incubated with Fc block and fixable viability stains (eBioscience) in FACS buffer for 20 min at 4 °C. Cells were washed with FACS buffer and incubated with MHC class I tetramer for 30 min at 4 °C. After incubation cells were washed with FACS buffer and were further incubated with the fluorochrome conjugated mAb cocktail. Antiviral CTL responses were quantified by staining with phycoerythrin (PE) labelled H-2D^b MHC class I tetramer loaded with the immunodominant influenza virus nuclear protein (NP) epitope NP(366-374) (ASNENMETM) (NIH tetramer core facility, USA) or OVA(257-264) (SIINFEKL (N4)) peptide (MBL, USA).

To measure antigen-specific cytokine release single cell preparations from lungs were stimulated in the presence of 10 μM NP(366-374) peptide and 1 μg/ml Brefeldin A for 6 h. Cells were stained for surface markers, fixed and permeabilized with the cytofix/cytoperm reagent (BD Biosciences, USA) and co-stained for intracellular cytokines and Granzyme-B.

**Infection with Listeria monocytogenes**. A total of 8–12-week old male and female mice were used for all experiments. Bacteria were grown in BHI medium to an OD₆₀₀ of 0.1 before each experiment. Mice were infected with a sublethal dose of $5 \times 10^6$ CFU attenuated (ΔactA) *listeria monocytogenes* expressing OVA(257-264)[49] by intravenous administration.

Adoptive transfer experiments: naive OT-I cells (if not otherwise stated) were sorted by Flow Cytometry from spleens and LN of mice from a respective genotype and 1,000 cells transferred intravenously on the day before infection. In all cotransfer experiments donor cells were mixed with 1*10⁶ carrier splenocytes of the same genotype as the recipient.

For characterization of transferred OT-I cells after transfer mice were bled on indicated dates and spleens were collected at final points of analysis. Cells were incubated with 10⁻⁷M N4 peptide (Genscript RP10611) for 3 h in the presence of Brefeldin A (1 μg/ml) in full cell culture medium. After surface staining, cells were fixed with 2% PFA for 20 min at 4 °C and permeabilized with BD Permwash + 1% FCS for 20 min at 4 °C, before intracellular Cytokine staining.

**Flow cytometry and monoclonal antibodies**. For cell surface staining single cell suspensions from tissues or cultured cells were prepared in FACS buffer containing 1x PBS, 1% FCS ± 2 mM EDTA (if not otherwise stated in the methods). All cells were blocked with Fcγ blocking antibody (24G2, BioXcell) and incubated with fixable cell viability dye eF780 (Thermo Fisher or BD) for 20 min at 4 °C.

For intracellular staining cells were fixed with BD Cytofix/Cytoperm (554722) or 2–4% PFA for 20 min at 4 °C. Cells were permeabilized with BD Permwash (554723) containing 1% FCS for 20 min at 4 °C. Intracellular staining was performed in BD Permwash containing 0.5% FCS and the intracellular antibody cocktail. Cytokine and Granzyme-B staining was performed for 1 h at 4 °C. Intracellular staining for transcription factors was performed for 2 h at RT. Staining for intracellular Notch1-IC was performed using fixation and permeabilization with Foxp3 transcription factor staining kit (Invitrogen; 00-5523-00). Intracellular staining was performed at room temperature in BD Foxp3 permeabilization buffer.

The following antibody clones were used in the flow cytometry experiments: CD4 (GK1.5,RM4-5;1:400), CD8 (53-6.7;1:400), CD45.1 (A20;1:100), CD45.2 (104;1:100), CD45 (30-F11;1:100), KLRG1 (2F1;1:400), CD127 (A7R34;1:100),

CD44 (IM7;1:400), CD25 (7D4, PC61;1:400), CD69 (H1.2F3;1:400), MHCII (M5/11415.2;1:400), TCRβ (H57-597;1:200), CD11c (cN418;1:400), F4/80 (BM8;1:400), NK1.1 (PK136;1:400), CD19 (6D5;1:400), Thy1.1 (OX7;1:400), TNF (MP6-XT22;1:200), IFN-γ (XMG1.2;1:400), IL2 (JES6-5H4;1:100), Granzyme-B (GB11;1:200), c-REL (1RELAH5;1:200), NFKB2 (EPR18756;1:200), IRF8 (V3GYWCH;1:100), NOTCH1-IC (mN1A;1:50), EOMES (Dan11mag;1:400), Cre (D7L7L;1:50). CD16/32 2.4G2 BioXcell BE0008 (1:2000, 0.5ug/ml). Secondary antibodies: polyclonal goat anti Rabbit (H + L) AF488 (A11034, Invitrogen;1:1000). Data was acquired using a Fortessa Flow Cytometer equipped with 355 nm, 405 nm, 488 nm, 561 nm and 640 nm lasers (Beckton Dickinson). Flow Cytometry data was analysed using FlowJo 10.6 software.

**In vitro culture and activation of T cells.** All in vitro and cell culture experiments were performed in IMDM culture medium (Gibco) supplemented with 10% FCS, GlutaMAX (Gibco), Penicillin /Streptomycin (Gibco) and 50 μM β-Mercaptoethanol.

Naïve CD8+ T cells were isolated from Spleen and Lymph Nodes using two rounds of negative depletion with Streptavidin DynaBeads (Thermo Fisher) using $1*10^8$ and $4*10^7$ beads per $1*10^8$ total splenocytes/LN cells. The following antibodies were used for the depletion cocktail: B220-Bio (RA3-6B3), CD4-Bio (GK1.5), CD11b-Bio (M1/70), CD11c-Bio (N418), CD19-Bio (1D3), CD44-Bio (IM7), CD105-Bio, F4/80-Bio (BM8), GR1-Bio (RB6-8C5), NK1.1-Bio (PK136), Ter119-Bio (Ter-119), γδTCR-Bio (GL3).

Cells were stimulated at a starting concentration of $5*10^5$ cells/ml, if not otherwise stated with 5 μg/ml plate bound anti-CD3 (145-2C11, BioXcell) and 1 μ/ml plate bound anti-CD28 (37.51, BioXcell). OT-I transgenic naïve CD8+ T cells were stimulated with (N4) SIINFEKL peptide at $10^{-10}$M. In some conditions recombinant mouse IL-2 (Peprotech) was added at 20 nM. For the analysis of proliferation, cells were labelled with cell trace violet (Thermo Fisher) or cell trace yellow (Thermo Fisher) at 10 μM final concentration for 6 min at 37 °C, before stimulation. Cell numbers per generation were enumerated using cell counting beads for flow cytometry (ACBP-50-10, Spherotech).

For the in vitro generation of CTL $1*10^6$ cells per ml of total splenocytes/LN cells from OT-I transgenic mice and cultured as follows: Day 0: total splenocytes were seeded at $0.5 \times 10^6$ cells per well in a 24-well plate and stimulated with 0.1 μM N4 peptide and 20 ng/ml recombinant murine interleukin-2 (IL-2) (PeproTech 212-12) and 2 ng/ml IL-12 (PeproTech 210-12) in 2 ml of Iscove's Modified Dulbecco's Medium (IMDM) (Thermo Scientific: 21980) supplemented with 10% FBS, 50 μM 2-mercaptoethanol (Thermo Scientific: 31350-010), 100 U/ml penicillin, and 100 μg/ml streptomycin (Thermo Scientific™: 15140). Day 2: All cells were transferred to a 15 ml falcon tube. Cells were centrifuged at ~300 g for 5 min at room temperature and washed twice in 5 ml of prewarmed DPBS (Dulbecco's phosphate-buffered saline) (Gibco: 14190250) and subsequently seeded at $0.25*10^6$ cells/ml in IMDM, supplemented with 20 ng/ml IL-2. On each subsequent day 0.5-$1*10^6$/ml cells were reseeded in IMDM, containing 20 ng/ml IL-2. On day 7 differentiated CTLs were used in subsequent experiments.

**Cytokine assays.** Mouse lL-2 secretion assay was performed following manufacturer's instructions (130-090-491; Miltenyi Biotech). In brief; $5*10^5$/ml purified naïve OT-I cells were activated with $10^{-10}$M N4 peptide in cell culture medium for the indicated timepoints. Cells were washed twice and incubated with capture reagent for 1 h at 37 °C while mixing the cells every 10 min. Cells were surface stained with antibody cocktail containing anti IL2-PE antibody.

**Cytotoxicity assays.** In vivo: Splenic leukocytes from naïve C57BL/6 mice (8–12 weeks old) were pulsed for 1 h with 10 μg/ml influenza virus NP(366-374) peptide and were subsequently labeled for 10 min at 37 °C with 5 μM CFSE (Invitrogen) (CFSE$^{high}$; loaded cells). Non-peptide loaded control cells were labelled with 0.5 μM CFSE$^{low}$. The two populations of cells were then mixed together in a 1:1 ratio and $6 \times 10^6$ cells were transferred intravenously into mice that had previously been infected with influenza PR8 ten days before, or transferred into uninfected control mice. Mice were killed 1 hour later and the percentage of peptide loaded target cells and peptide non-loaded cells was quantified by flow cytometry.

In vitro: OT-I CTL were cultured in the presence of N4 peptide -pulsed (target), and non-pulsed (non-target) EL4 mouse cell line. Target cells were pulsed with $10^{-7}$M N4 peptide for 30 min at 37 °C and washed 1x with ice cold FBS and 2x with cell culture medium. Target and non-target cells were marked with CTY and CTV respectively at 10 μM final concentration. Killing experiments were performed at different ratios of target versus OT-I cells for 3 h at 37 °C.

The percentage of target cell killing was determined as: 100 − {(percentage of peptide-pulsed targets in infected recipients/percentage of un-pulsed targets in infected recipients)/ (percentage of peptide-pulsed targets in naïve recipients/un-pulsed targets in naïve recipients) X 100}.

**CrispR Cas9 experiments and sgRNA cloning.** MSCV_hU6_BbsI-ccdB-BbsI_iScaffold_mPGK_puro-2A-Thy1.1 (MSCV_sgRNA_puro-Thy1.1) was generated from vectors MIGRI and PKLV2_sgRNA_puro-Thy1.1. For individual sgRNAs two 24nt oligonucleotides were annealed and ligated (T4) into a BbsI

linearized MSCV_sgRNA_puro-Thy1.1 vector. Pairwise sgRNAs were initially independently cloned alongside either the hU6 or mU6 promoter then amplified by PCR and ligated in tandem into the destination vector. Plat-E cells at $0.3*10^6$ cells/well of 12-well plate were seeded and cultured overnight before transfection with 100 μl OptiMEM (Gibco), 3 μl TransIT-293, 0.8 μg transfer vector, 0.2 μg packaging pCL-ECO. Retrovirus was harvested and frozen from transfected plat-E cells over the subsequent 72 h.

For CRISPR/Cas9-gene editing experiments $1*10^6$ cells per ml total splenocytes/LN cells from OT-I mice expressing Cas9 endonuclease (OT-I+; CD4$^{cre}$; Rosa26lsl; Cas9-GFP) were activated with $10^{-7}$M N4 peptide in the presence of 20 nM IL-2 and 2 nM IL-12 for 24 h. Plasmids encoding single guide RNAs (sgRNAs) were introduced by retroviral transduction into activated OT-I cells by centrifugation at (1000 x g) for 45 min at 32 °C. Following transduction OT-I cells were maintained with exogenous IL2 for 24 h. Transduced cells were isolated by sorting cells expressing Thy1.1 (CD90.1) transduction marker. Cells were maintained for a further four days with exogenous IL-2 to generate CTLs.

To test for the efficiency of CRISPR/Cas9-induced knock out of Zfp36 and Zfp36l1 CTLs were stimulated for 3 h with $10^{-7}$ M N4 peptide, and expression of Zfp36 family members was detected by Western blotting. Guide RNA combinations to target individual ZFP36 family members were used as follows:

**Non-targeting guides.** GGTAGAGTCGCGAACGCTAC;
GATGCGGACCCACGTTAAGC;
GGCATGGAGTCAACGTCCGC
Guides targeting both Zfp36 and Zfp36l1
GAGCGGGCGTTGTCGCTACG + GGAGTGACCGAGTGCCTGCG;
GAGCGGGCGTTGTCGCTACG + GTTGAGCATCTTGTTACCCT;
GATGACCTGTCATCCGACCA + GTTGAGCATCTTGTTACCCT;
For electroporation of Cas9 Ribonucleoproteins naïve OT-I T cells were isolated using the StemCell EasySep Mouse Naïve CD8+ T Cell Isolation Kit (StemCell; 19853). Isolated cells were electroporated with complexes of Cas9 and control or IL2 targeting gRNA (all from IDT) in OptiMEM medium (Gibco) using the NEPA21 electroporator (Nepagene). After electroporation cells were cultured for 24 h in RPMI medium (Gibco) containing 10% FCS and 10 ng/ml recombinant IL7 (Peprotech; 217-17), before transfer into recipient mice. The following guide sequence (IDT) was used to target the the indicated genomic sequence of the Il2 gene: AAGATGAACTTGGACCTCTG.

**Western Blotting.** Whole cell lysates were prepared using 1x RIPA buffer [SDS 0.1 % (w/v), Sodium deoxycholate 0.4% (w/v), NP-40 0.5% (v/v), 100 mM NaCl, 50 mM Tris-HCl, pH 7.4], supplemented with 1:100 protease inhibitor cocktail (Sigma P8340), phosphatase inhibitor cocktail III (Sigma P004), and 2 U/ml Benzonase (9025-65-4). Protein lysate concentration was determined using Pierce BCA protein assay kit (23225). Lysates were denatured for 5 min at 97 °C with Laemmli buffer containing 5% β2-mercaptoethanol, and the equivalent of 30 μg of total protein was loaded per lane. Samples were resolved by 10% SDS-PAGE and transferred to nitrocellulose membrane using iBlot2 transfer device (IB21001). The membrane was probed with anti-ZFP36 mouse monoclonal (Origene #OTI3D10) (2 μg/ml) and anti-ZFP36L1 polyclonal (CST #BRF1/2) (32 ng/ml) antibodies. Proteins were detected by incubating membrane with anti-Mouse IgG IRDye800CM (Licor #926-32210) and anti-Rabbit IgG IRDye680RD (Licor #925-68071), and scanning the membrane with Licor Odyssey CLx using standard methods. GAPDH was detected using rabbit monoclonal anti-GAPDH (CTS #5174) at 1:1,000 and anti-Rabbit IgG IRDye680RD. Image analysis was conducted using ImageStudio Lite version 5.2, and normalized protein signal was calculated using standard methods.

**Real time PCR analysis.** RNA was isolated using TRIzol(Ambion;15596018)/Chloroform extraction. Genomic DNA was removed using DNAseI and RNA was cleaned up by column purification (Qiagen; 74004). RNA was transcribed to cDNA with RevertAid first strand cDNA (Thermofisher; K1622). Target gene expression was detected by qPCR using TaqMan probes (Thermofisher; 4304437; 4331182 Mm99999222_m1 mouse IL2) and normalized to the expression of RPL32 (Thermofisher; 4351372 Mm07306626_gH RPL32). For the detection of unprocessed IL2 mRNA, custom primers were designed across the boundary of Exon 1 and Intron 1 (FW: AGCTGTTGATGGACCTACA, REV: CACAGCCTTTGG-CAAGAAA). Expression was detected using Platinum SYBR green reagent (Invitrogen; 11744-500) and was normalized to the expression of RPL32 (FW: ATCAGGCACCAGTCAGACCGAT, REV: GTTGCTCCCATAACCGATGTTGG.

**ZFP36L1 iCLIP analysis.** A total of $25*10^6$ in vitro-generated CTL from three biological replicates were crosslinked at 150 mJ/cm$^2$ with Stratalinker 2400 with 245 nm radiation, and snap frozen on dry ice. Cells were then lysed and sonicated, and treated with RNase I (0.03 U/ml) [AM2294] and Turbo DNase (2 U/ml) [AM2238] at 37 °C for 3 min at 1,100 rpm. Anti-ZFP36L1 polyclonal antibody (2 μg/ml) [ABN-192] was pre-coupled to protein A/G Dynabeads [Invitrogen 88802], and incubated with lysates under rotation overnight at 4 °C. ZFP36L1-RNA complexes were magnetically separated, and RNA was de-phosphorylated using FastAP Alkaline phosphatase [Invitrogen EF0651] for 40 min at 37 °C at

# ARTICLE

1100 rpm. IRDye-linked pre-adenylated adaptors were ligated to RNA using T4 RNA ligase I (M0437M), in the presence of 11.25% PEG8000, at room temperature for 75 min. Non-ligated adaptors were removed using RecJF 5′-3′ exonuclease (M0264S) for 1 h at 30 °C at 1100 rpm, and RNA was de-adenylaed using NEB de-adenylase (M0331S) for 30 min at 37 °C at 1100 rpm. ZFP36L1-RNA complexes were removed from magnetic beads by re-suspending in NuPAGE LDS buffer (sup with 100 mM DTT), and heating to 70 °C for 1 min and shaking at 1100 rpm. The supernatant was loaded onto pre-cast 4-12% NuPAGE Bis-Tris gel [Invitrogen] and resolved by running for 65 min at 180 V with MOPS buffer (NP0001). Complexes were transferred to nitrocellulose membrane using XCell II Blot module [Invitrogen] for 1 h at 30 V using standard methodology. Nitrocellulose membrane was scanned using Licor Odyssey CLx and ZFP36L1-RNA complexes excised from membrane by cutting 35–55 kDa above expected running weight of ZFP36L1. Protein was digested by incubating membrane at 50 °C for 1 h with proteinase K (Merck) diluted in PK-SDS buffer. Supernatant was mixed with phenol:chloroform:isoamyl alcohol and transferred to 2 ml phase-lock gel tube and spun for 5 min at 17,000 x g at room temperature. Chloroform was added to the top phase and samples were centrifuged for 5 min at 17,000 x g at 4 °C. Aqueous phase was isolated and RNA precipitated over night at −20 °C with 100% Ethanol, 1 M NaCl and Glycoblue (0.1% v/v) [AM9515]. Reverse transcription was conducted using Superscript IV [Invitrogen 18090010] and 1 pmol/µl irCLIP_ddRT primers featuring 5nt unique molecular identifiers. cDNA was purified using AMPure XP beads [Beckman Coulter A63881] and circularized in the presence of Beatine (1 M) using CircLigase II [Lucigen CL9021K]. PCR amplification was conducted using P5 and P3 solexa primers [Illumina] and Phusion HF master mix [NEB M0536S]. PCR products were run on 6% TBE gel and DNA in the range of 145-400 nt isolated using standard methodology. iCLIP libraries were multiplexed and sequenced using HiSeq2500-RapidRun (100 bp Single End).

**ZFP36L1 iCLIP mapping**. iCLIP data was mapped to GRCm38 mouse genome assembly using bowtie2. Barcoded adaptors of iCLIP sequenced reads were removed before mapping, crosslink sites were defined by the nucleotide preceding iCLIP cDNAs using Genialis iMaps web server: https://imaps.genialis.com/iclip.

**Data and statistical analysis**. HITS-CLIP datasets for ZFP36 in CD4+ T cells following 4 h or 72 h activation were obtained from GSE96074[20]. This data, together with iCLIP data, were processed using the Genialis iMaps web server (https://imaps.genialis.com/iclip). Data was deduplicated based on the random barcodes, trimmed with Cutadapt[50], and mapped to GRCm38 mouse genome assembly using STAR[51]. Significant crosslink sites, defined by the nucleotide preceding iCLIP or HITS-CLIP cDNAs, were identified using the iCount pipeline (https://icount.readthedocs.io/en/latest/index.html).

For CLIP target gene identification, target transcripts were filtered for the presence of identical significant crosslink sites in their 3′UTRs in at least three out of four or five (for pan-ZFP36) and two out of three (for ZFP36L1) biological replicates. Unique and common targets identified from the two datasets were then unified and taken as our final list of high confidence ZFP36/L1 target genes.

RNA-seq data for naïve and in-vitro activated CD8+ T cells were obtained from GSE77857[27]. Data were trimmed using Trim Galore (https://www.bioinformatics.babraham.ac.uk/projects/trim_galore), mapped to the GRCm38 mouse genome build using Hisat2[52], taking into account known splice sites from the Ensembl Mus_musculus.GRCm38.90 annotation release. Raw read counts over mRNA features from the same annotation release were quantified using Seqmonk (https://www.bioinformatics.babraham.ac.uk/projects/seqmonk). Differentially expressed genes at 6 h and 18 h after CD8+ T cells activation were identified using DESeq2 analysis with default parameters for each time point relative to 0 h, and were selected for adj. $p$ value < =0.05 and $1.3 < \log 2\,FC < -1.3$ (using 'normal' log2 fold change shrinkage). Genes with a lower baseMean expression than 30 normalized read counts were excluded from analysis.

For analysis of binding site conservation, a 70nt sequence around the identified CLIP binding site was analyzed for conservation among vertebrates using the UCSC genome browser. Downstream analysis of CLIP data was performed using R v4.0.4.

**Reporting summary**. Further information on research design is available in the Nature Research Reporting Summary linked to this article.

## Data availability

All data generated and analyzed during this study are included in this article and its supplementary information, or have been made available in public repositories as follows: Sequencing data from ZFP36L1 iCLIP experiments performed in this manuscript is publicly available on: GEO: GSE176313. Previously published datasets are available on GEO under the accessions GSE77857 (RNA-seq) and GSE96074 (HITS-CLIP for ZFP36). Source Data is provided as a Source Data file for data sets presented in this study.

Statistical analysis was performed with Graph Pad Prism 8.1.2 and R v4.0.4.

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

## Acknowledgements

We thank the Babraham Institute Biological Support Unit, Flow Cytometry and Kirsty Bates, Anne Segonds-Pichon and the Bioinformatics Facilities for assistance; Elisa Monzon-Casanova, Adrian Liston, Sarah Ross and Arianne Richard for comments on the manuscript. This study was supported by funding from the Biotechnology and Biological Sciences Research Council (BBSRC) (BBS/E/B/000C0407; BBS/E/B/000C0428; the BBSRC Core Capability Grant to the Babraham Institute; and a Wellcome Investigator award (200823/Z/16/Z) to M.T. FS was supported by European Molecular Biology Organization (EMBO) Long-Term Fellowship (ALTF 880-2018). T.J.M. was supported by the BBSRC Cambridge doctoral training partnership. V. D′A. was supported by the Cambridge Trust.

## Author contributions

G.P. conceptualization; methodology; investigation; validation; formal analysis; visualisation; writing original draft preparation; T.J. M. conceptualization; methodology; investigation; validation; formal analysis; visualisation; writing review and editing; K.C. conceptualization methodology; investigation; visualisation; writing review and editing; S.E.B. conceptualization; resources; methodology; writing review and editing; V. D′A. investigation; writing review and editing; L.M and O.G. software; formal analysis; investigation; visualisation; writing review and editing; A.S. and D.J.T. and F.S. methodology; writing review and editing; P.D.K. conceptualization; resources; writing review and editing; M.T. Conceptualisation, supervision, funding acquisition, writing review and editing.

## Competing interests

The authors declare no competing interests.
