## [Peer Review File · Nature Communications]

The timing of differentiation and potency of CD8 effector function is set by RNA binding proteinsREVIEWER COMMENTS

Reviewer #1 (Remarks to the Author):

This manuscript by Petkau et al. investigates the regulation of effector cell generation and function by the RNA binding proteins ZFP36 and ZFP36L1. The regulation was shown to be cell intrinsic. The absence of ZFP36 and ZFP36L1 results in CD8 T cells with increased effector function. The effects of the RNA BP are due to binding of NFkB, IRF8, and NOTCH and through limiting IL-2 production. The data are clearly presented and well described. These data are an important contribution to our understanding of T cell activation and differentiation as they shed new light on the requirement for CD28 signaling and how the system is calibrated to enforce this requirement. Following are suggestions for changes that would increase clarity and provide additional insights.

1. There appears to be variability in the WT groups in Figure 1a and b. The error bars look relatively tight, so it seems to be an assay-to-assay difference. It would allow more rigorous interpretation of the effects of the two KOs if they were in a side-by-side assay.

2. How does the number of virus-specific cells change in the lung over time? It's interesting that a difference in clearance is not observed until d10 p.i. This presumably reflects later entry of high numbers of antigen-specific cells.

3. The data in figure 1g appear to be gated on total CD8 cells. Day 4 would be very early for readily detectable virus-specific cells. Thus, the increase at that point may reflect other responses/differentiation processes rather than infection. The authors should comment on this.

4. The day of analysis moves around a bit in Figure 1, panel h is d8, others are d10. Consistency would be advantageous.

5. Do the authors know if the sensitivity of the effector cell is altered in the KOs?

6. The lower number of antigen-specific cells in the lung is a bit surprising. Is this reflected in the draining lymph node or is it a trafficking/survival issue?

7. What are the data that support the conclusion that there is a delay in the onset of effector functions (last sentence pg4)?

8. Were all cytokines dependent on the late presence of ZFP36 and ZFP26L1 or just TNF?

9. It would be useful to know the number of cells recovered in each of the cultures in Fig 3.

10. Did the authors costain for eomes, granzyme B and division?

11. I disagree that the increased proliferation in Fig 7a,b demonstrates a reduced activation threshold. This conclusion would require stimulation with titrated anti-CD3.

12. Is the statement regarding accumulation of more cells per generation meant to suggest that for each cell that enters division there are fewer cells remaining by division 4 or simply that there are fewer cells that made it to division 4. The former would suggest differential survival.

13. What happens to ZFP36 and ZFP36L1 expression in WT cells stimulated with anti-CD3 and CD28?

14. Why do the authors think the individual effector functions are differentially regulated at early versus late times (Fig.3)? It would be great to work this into the model.

15. It would be helpful to discuss this pathway in the context of a normal immune response. This

appears to be the brakes on the system to ensure the need for a stimulus that results in DC maturation. Some of this information is in the figure legend of SF6, but more in the body of the discussion would be useful.

Minor points:

1. It would be helpful to note in the text of the results the day p.i. or stimulation that the assay is performed as it is not consistent within the figures.
2. Please add % numerical values to data in F3d and f.
3. The authors should state "data not shown" after the finding where NFkB and IRF8 were measured after IL-2 was added.
4. Delete "to" in first line of discussion.
5. Please add reference for the Roquin paper described in Discussion.
6. Space needed "andTBET" in Discussion.
7. The title for SF1 is incorrect as both flu and Lm are used. Consider dividing into two figures.
8. Gamma symbol lost in legend of SF1.
9. Please add the group explanations for I,II,III,IV, and V to legend of SF3.

Reviewer #2 (Remarks to the Author):

This study shows that the RNA binding proteins ZFP36 and ZFP36L1 control the extent to which naïve CD8+ T cells differentiate into effector cells that limit immunopathology and viral load, and eliminate antigen-loaded target cells in vivo. Mice doubly deficient (dKO) for these RBP lost less weight than WT mice after primary infection with the highly pathogenic PR8 strain of IAV, and were protected against lethal PR8 infection. Similar results were seen when WT mice were injected with small numbers of WT or dKO OT-I cells, demonstrating that effects were CD8+ T cell-intrinsic. By d10, the response of dKO cells was moderately skewed towards an SLEC phenotype, (although most of the surviving OT-I cells appeared to express neither CD127 nor KLRG-1). OT-I cells that could only express ZFP36 and ZFP36L1 during the first 24 hours after stimulation behaved like WT rather than dKO cells, consistent with the hypothesis that ZFP36 and ZFP36L1 act early in CD8+ T cell differentiation. Expression of transcription factors important for effector function was also enhanced in dKO cells. Finally, WT OT-I cells were significantly more dependent on CD28 costimulation compared with dKO OT-I cells.

Based on these results, the authors discuss how ZFP36 and ZFP36L1 limit CD8+ T cell activation and differentiation into effector cells. Rather than just limiting effector cell development, however, ZFP36 and ZFP36L1 might be critically important for the differentiation of naïve CD8+ T cells into MPEC rather than SLEC. Response skewed towards SLEC would be highly effective at viral clearance and limiting immunopathology during primary infection, but at the expense of memory formation and protection against viral challenge. Do dKO OT-I cells make significantly impaired memory responses compared with WT OT-I cells to a second IAV infection? This would reframe the discussion away from limiting effector differentiation, towards an important role in memory formation.

One intriguing result: adoptive transfer of 200 dKO OT-I cells had a dominant effect on the endogenous CD8+ T cell response, as shown in Fig.S1a,b. Transfer of 200 OT-I cells should result in engraftment of about 20 OT-I cells (assuming ~10% engraftment success rate), rather fewer than the number of naïve endogenous SIINFEKL-specific CD8+ T cells (70 - 600; reviewed in Jenkins et al, 2012, J. Immunol. 188: 4135). Fig. S1a, b, show similar numbers of endogenous SIINFEKL-specific CD8+ T cells and WT OT-I cells in the spleen and lung 10 days after infection with IAV. In contrast, the number of endogenous cells is reduced almost 10-fold when dKO OT-I cells were transferred. Perhaps these data could be incorporated into Fig. 2? They're certainly consistent with the idea of a rapid effector response by dKO cells that could reduce viral infection loads, as mentioned in the discussion.

There's some concern about the very exciting culture conditions used to generate CTL in vitro (Fig.

3). CD8+ T cells expand extremely rapidly after SIINFEKL activation of OT-I spleen cells, differentiate into highly potent CTL even in the absence of IL-12, and must be diluted very frequently to maintain optimal viability. Cells grown in very high dose IL-2 (20nM = ~300ng/mL) without IL-7 or IL-15 generally don't revert to resting cells, are highly cytotoxic, and are extremely susceptible to apoptosis. The limited GzmB expression and relatively poor cytotoxicity (less than 50% lysis at 1:1 E:T) of WT OT-I cells after 7 days culture was surprising. More information on culture conditions and cell viability would be helpful.

Minor points:

The diagram in Fig. 2a suggests that ZFP36 and ZFP36L1 are also expressed during the 3h in vitro cytotoxicity assay - has this been shown?

Could the flow data shown in Fig. 2b-f could be shown as un-smoothed pseudocolor plots?

Figures showing gating strategies and appropriate controls for cytokine secretion assays (see Figs. 2b-f; Fig 3 d,f; Fig. 6g) should be included in the Supplementary Data (Fig. 5 looks fine). This is especially important for cells grown for

Since granzyme B can be stored in granules, it would be very helpful to include intracellular cytokine staining of unstimulated as well as peptide-restimulated cells in Fig. 3d, or show these as supplementary figures.

The data in Fig. 4a were presumably generated from peptide-stimulated naïve OT-I cells, not CTL as stated (p7, second paragraph, line 9).

The data points for cells stimulated with CD3 without CD28 (mentioned in the text on p11) should be added to Fig 7b.

Reviewer #3 (Remarks to the Author):

In this report Petkau et al. focuses on the role(s) of RNA binding proteins (RBPs) Zfp36 and Zfp36l1 during naive CD8 T cell activation and the response to Flu infection in mice. Analysis of mice in which genes encoding Zfp36 or Zfp36l1 were conditionally inactivated individually or in conjunction before naive T cells develop using the Cd4-Cre transgene (deletes in double-positive thymocytes) showed that elimination of both genes (double knockout [dKO]), but not either gene alone, increased viral clearance and decreased morbidity (less weight loss). Improved viral control in dKO mice is associated with increased frequencies of cells that expressed an effector cell phenotype (Klrg1hi, and increased Gzmb, IFNg, TNF expression) and increased in vivo killing function, and appears to be CD8 T cell intrinsic. The requirements for these RBPs appear to be early after TCR-activation because their inactivation after in vitro activation of naive CD8 T cells using CRISPR-Cas methods exhibited a less profound phenotype compared to Cre-mediated disruption. Multiple mRNA targets were identified using HITS-CLIP approach, and comparisons to mRNAs whose expression changes following initial activation of naive CD8 T cells. The protein-level expression of several of these targets (Il2, Nfkb2, Rel, Irf8 and Notch1) were found to be increased in cells lacking Zfp36 and Zfp36l1. The authors' main overarching conclusion is that the Zfp36 family of RNA binding proteins limit the speed and quality of the CD8 T cell response.

The molecular mechanisms that are involved in the initial activation and differentiation of naive CD8 T cells is a significant and interesting topic, and clearly have relevance for understanding how to control viral infections. The HITS-CLIP dataset in recently stimulated naive CD8 T cells appears to be high quality and will be a resource that should be of interest to the community. Overall, the manuscript is written clearly and the quality of the experiments and their presentation is high. However, a several issues limit the strength of the biological and mechanistic insights that can be currently drawn.

Major points:

1. The finding in this study that Zfp36/Zfp361 also enhances antiviral immunity, in this case against influenza A infection, is important, but has less general impact because previous studies have reported that Zfp36-deficiency enhances clearance of viral infection in other models (Moore et al. *Elife*, 2018).

2. This report provides some evidence that Zfp36/Zfp361 regulates Il2 mRNA, which is a different observation than previous studies. However, the extent of the mechanistic insight as to how this occurs limits the interpretations that can be drawn at present. The authors postulate that Il2 dysregulation in the absence of Zfp36/Zfp361 results from direct (Il2 mRNA binding) and indirect (Nfkb mRNA binding) because of NF- κ B's appreciated role in promoting Il2 transcription. However, the relative contribution of NF- κ B and direct action of Zfp36/Zfp361 on IL-2 message is not delineated and whether one or both explains the difference in IL-2 expression between WT and dKO cells is presumed, but not demonstrated. Is Il2 transcription affected, or is IL-2 mRNA half-life different, and do these changes depend on the cis-elements in Nfkb2 and Il2 mRNAs in the absence/presence of Zfp36/Zfp361? Are there differences in NF- κ B-controlled regulatory elements in the Il2 locus in WT versus dKO cells? Can these changes be linked to the in vivo phenotype of the dKO CD8 T cells during infection?

3. The nature of the interpretations in this study turn on the idea that Zfp36/Zfp361 are mediating their functions following TCR stimulation of naive cells. However, naive cells from the dKO mice do not seem equivalent to WT cells before activation. NF- κ B2 and cRel protein expression both appear to be greater in dKO compared to WT cells before activation (Fig. 6e and f). Are there any differences in the phenotypes of cells isolated from naive WT and dKO mice before activation? A careful comparison of T cells from WT and dKO mice is important to demonstrate if cells from both backgrounds are on equal footing before activation, or to at least document what is already different before initial activation.

4. The authors' interpretation that the improved clearance of flu infection in mice lacking Zfp36/Zfp361 is due to enhanced effector functions in responding CD8 T cells. The kinetics of the infection are different in WT mice compared the dKO mice, which undoubtedly will have affected the number and phenotype of the responding cells. This makes apples-to-apples comparisons between different groups of mice challenging, and some of the observed differences in the frequencies of phenotypic populations of CD8 T cells could relate to differences in the infection kinetics in WT versus dKO mice. In addition, to what extent the differences in the flu response are due to altered effector function capability, or differences in effector cell accumulation in the absence of Zfp36/Zfp361 is not clearly resolved. The data presented regarding effector cell numbers in infected mice is confusing. Supplemental Fig 1a and 1b appears to be contradictory with the text. The figure indicates that there is a reduction in Ova-specific "recipient" cells, while the number of "donor" (OT-I) cells is largely unchanged between WT and dKO groups. However, the text states that: "Consistent with our studies in intact mice there are fewer dKO OT-I cells in the lung and the spleen at day 10 of infection than in mice with transferred WT OT-I cells (Supplementary figure 1a, b)". Furthermore, it is difficult to reconcile the conclusion that there are fewer effector cells in dKO mice compared to WT mice, but there is increased target cell killing in dKO recipients during the in vivo lysis assay (Fig 1a), because the killing of targets in vivo is thought to proceed according to mass action, and even memory cells that have virtually undetectable perforin/granzyme B protein and killing activity in vitro (i.e., central memory cells) kill almost as rapidly as effector cells from the peak LCMV response when assayed in vivo (Barber et al. 2003, *J. Immunol*). Altogether, these issues bring into question whether enhanced effector functions in dKO cells is the explanation of the improved viral control, or if it could relate to more rapid accumulation of effector cells. The authors' observation that Zfp36/Zfp361 bound to the key cell cycle regulators Cdk1 and Cdk6 could be an indication that the rate at which responding cells enter the response is accelerated in the absence of Zfp36/Zfp361, but this is not explored in terms of cell numbers early in the response.

Specific comments/suggestions:

1. This sentence in the results is confusing: "To identify direct targets of the RBP we performed ZFP36L1 "individual-nucleotide resolution UV crosslinking and immunoprecipitation" (iCLIP) on OT-I CTLs stimulated with peptide for three hours and combined this with "high-throughput sequencing of RNA isolated by crosslinking immunoprecipitation" (HITS-CLIP) from CD4 T cells activated for 4 hours". What was actually done here?
2. The rationale for the selection of experiments in Fig. 4 seems incomplete. An important dataset that seems to be missing is one in which genes whose expression is altered in the absence of Zfp36/Zfp36l1. Do these RNA binding proteins universally negatively regulate the half-life of their bound targets, or are there different modes of action depending on which mRNAs or which elements in those RNAs they bind?
3. Ideally, another Cre-driver that acts after the thymus seems to be an important complement to the study. In addition, all comparisons in the study use mice and cells with Zfp36/Zfp36l1 "floxed" alleles without CD4-Cre transgene as the negative control. A formal comparison to CD4-Cre+ cells/mice that have WT Zfp36/Zfp36l1 alleles is also an important negative control, because Cre can have deleterious effects, especially in thymocytes.
4. Figure 4b. What does the red dotted line indicate?
5. Are there differences in IL-2 regulated genes in Zfp36/Zfp36l1-deficient CD8 T cells as a consequence of increased or accelerated IL-2 expression?
6. More clearly delineating in the text the situations in which more rapid gene expression kinetics versus qualitative changes in gene expression are evident would provide additional clarity to readers.

Dear Reviewers,

We would like to thank you for your constructive and helpful comments and suggestions on our manuscript "The timing of differentiation and potency of CD8 effector function is set by RNA binding proteins". We have addressed all comments and concerns raised by the reviewers by clarification and the addition of a substantial amount of additional data. We believe that by addressing the points stressed by the reviewers we were able to substantially improve the manuscript and deepen our understanding of the mechanisms controlled by ZFP36 and ZFP36L1 in order to limit T cell activation and effector function.

In the following we provide a point-by-point response to the (bold/italicized) comments of the reviewers.

Response to comments of Reviewer 1: This manuscript by Petkau et al. investigates the regulation of effector cell generation and function by the RNA binding proteins ZFP36 and ZFP36L1. The regulation was shown to be cell intrinsic. The absence of ZFP36 and ZFP36L1 results in CD8 T cells with increased effector function. The effects of the RNA BP are due to binding of NFkB, IRF8, and NOTCH and through limiting IL-2 production. The data are clearly presented and well described. These data are an important contribution to our understanding of T cell activation and differentiation as they shed new light on the requirement for CD28 signaling and how the system is calibrated to enforce this requirement. Following are suggestions for changes that would increase clarity and provide additional insights.

Thank you very much for your insightful comments and careful reading of the manuscript.

1. There appears to be variability in the WT groups in Figure 1a and b. The error bars look relatively tight, so it seems to be an assay-to-assay difference. It would allow more rigorous interpretation of the effects of the two Kos if they were in a side-by-side assay.

We acknowledge there is variability between experiments. In the manuscript, we are only making the comparison between the indicated conditional mutant and its littermate control group. This is the key comparison as these mice will have shared environmental history which is a significant variable for these infections. Here, we have focussed a more detailed study of mice where both RBPs are absent because we observed the strongest effects on body weight loss in the double KO. We do not compare the body weight loss of the dKO mice with that of the single RBP KO. We agree with the reviewer that it would be ideal to perform these experiments where all the genetically modified mice are infected and analyzed together, but this will exceed 30 mice per experiment and present a significant technical challenge to perform consistent infection and subsequent measurements. We want to emphasise that our result cannot be taken to indicate that there will be no phenotype in the single knockouts if they are challenged in different ways.

2 . How does the number of virus-specific cells change in the lung over time? It's interesting that a difference in clearance is not observed until d10 p.i. This presumably reflects later entry of high numbers of antigen-specific cells.

We have detected the presence of NP-specific CD8 T cells at day four in lung tissue. At this time, the numbers and frequencies are the same between WT and dKO. Also, on days 7 and 8, where we observe a massive increase in NP-specific CD8 T cells, the numbers between the WT and dKO in lung tissue are not different. Thus, only at day 10 do we observe a reduction in NP-specific CD8 T cells in the dKO and this is only by 50%. This may reflect an earlier contraction of dKO cells in the lung tissue. We have added this data into the main Figure 1 (f, g).

3. The data in figure 1g appear to be gated on total CD8 cells.

The reviewer is correct to highlight that data in figure 1g (now 1i) is gated on total CD8 T cells but that the accompanying text states incorrectly antigen specific T cells. We apologize and have corrected the statement (line 118).

Day 4 would be very early for readily detectable virus-specific cells. Thus, the increase at that point may reflect other responses/differentiation processes rather than infection. The authors should comment on this.

Previous studies have found that low numbers of influenza specific cells can be found in the lungs as early as day 4 (<https://www.ncbi.nlm.nih.gov/pmc/articles/PMC2762786/>). Our own data is consistent with this (*now included in Figure 1 in response to question no 2*) and also shows readily detectable NP-specific CD8 T cells in lungs on day 4 post infection. However, the number of influenza specific cells (when assayed for a single antigen specificity) are often underestimated during influenza infection when assessed by tetramer staining, especially in the early response (<https://doi.org/10.4049/jimmunol.174.9.5332>). This can be due to the inability of tetramers to detect low affinity CD8 T cells which contribute early to the response. GranzymeB expression has been shown to be a good measure of the overall CTL pool responding to the infection, especially early in the response (<https://doi.org/10.4049/jimmunol.174.9.5332>). We have clarified this in the results text (line 121).

4. The day of analysis moves around a bit in Figure 1, panel h is d8, others are d10. Consistency would be advantageous.

We thank the reviewer for highlighting this. The panel in figure 1h is showing data from day 10 post infection and not day 8. We have corrected this error in the corresponding figure legend.

5. Do the authors know if the sensitivity of the effector cell is altered in the KOs?

We have tested if the dKO CTLs have an altered sensitivity to antigen by performing a CTL killing assay with target cells pulsed with SIIVFEKL (V4) a low affinity variant of SIINFEKL (N4). We found that CTLs deficient for ZFP36 and ZFP36L1 killed target cells presenting low affinity antigen more efficiently than control CTL suggesting that the dKO CTLs are more sensitive to antigen. We have added this new data into a new supplementary figure (SF4c) and made references to these findings in the results text body (line 194).

6. The lower number of antigen-specific cells in the lung is a bit surprising. Is this reflected in the draining lymph node or is it a trafficking/survival issue?

The lower number of antigen-specific cells in the lung was only evident at day 10 as shown in revised figure 1. The reduced numbers of dKO CTLs are likely to be a reflection of a slightly earlier onset of the contraction phase which would be expected if antigen is cleared more rapidly. This observation is not necessarily surprising and we have not pursued this as it would require a detailed and expensive study. A very recent publication highlights biphasic behaviour of CD8 T cells during the course of the IAV infection indicating a non-linear correlation between T cell numbers and viral titres in the lung (<https://elifesciences.org/articles/68864>). We have added a paragraph to the discussion section addressing this and add the cited research as a new reference no.41 (line 415).

7. What are the data that support the conclusion that there is a delay in the onset of effector functions (last sentence pg4)?

Our intended meaning in the sentence being referred to “*Thus, in model viral and bacterial infections, Zfp36 and Zfp36l1 act within CD8 T cells to delay the onset and limit the magnitude of acquired effector functions*” is that when *Zfp36* and *Zfp36l1* are present the acquisition of effector function and thus differentiation into effector cells is delayed.

This is based on the finding that in two different infection models the dKO CD8 T cells more quickly acquire a SLEC phenotype and express higher levels of GranzymeB earlier than *Zfp36/Zfp36l1*-sufficient cells. In the listeria infection model where we are able to follow SLEC vs MPEC formation over time in the same mouse we observe that the accumulation of SLEC from dKO CD8 T cells is more rapid (SF2b). These observations, together with our *in vitro* observation of a more rapid acquisition of GranzymeB (already evident on day 3 after activation) (shown in figure 6) by activated naïve dKO CD8 T cells (Fig.6g), leads us to conclude that ZFP36 and ZFP36L1 delay the acquisition of an effector phenotype.

8. Were all cytokines dependent on the late presence of ZFP36 and ZFP26L1 or just TNF?

Besides TNF we have also assessed IL-2 and IFN γ expression in the CTLs. When we stimulated WT and dKO OT-I CTLs with N4 peptide we did not observe differences in IFN γ production. While CTLs produce only low levels of IL2, we did find an increase in the frequency of IL2 producing dKO CTL as compared to WT controls, but little, if any, difference in the amount of cytokine produced per cell. We have added the data to SF4e, f.

9. It would be useful to know the number of cells recovered in each of the cultures in Fig 3.

We have added the CTL numbers recovered during the killing assay in figure 3 into supplementary information (SF4b). We did not observe differences between WT CTL and dKO CTL numbers recovered. We made a comment on this in the results section (line 191).

10. Did the authors costain for eomes, granzyme B and division?

In the experiments presented in Figure 3 we did not co-stain for EOMES, Granzyme B and cell division. We find that labelling previously cultured cells with cell-trace dyes does not yield readily interpretable results, presumably because the cells are not uniformly labelled at the outset due to the much greater variation in cell size than is found with naive lymphocytes. Moreover, the cells in culture are not synchronised with respect to the stage of the cell division cycle.

11. I disagree that the increased proliferation in Fig 7a,b demonstrates a reduced activation threshold. This conclusion would require stimulation with titrated anti-CD3.

The reviewer is correct. We have now monitored proliferation of cell trace labelled WT and dKO naïve CD8 T cells in response to varying doses of plate bound anti-CD3 antibodies. We find that dKO cells respond more vigorously with proliferation and accumulate more cells per division as compared to their WT counterparts at each dose of anti-CD3. These new results and the new data with low affinity peptide (*response to point 5 above*) show that ZFP36 and ZFP36L1 raise the activation threshold of CD8 T cells. We have added this data into the main Figure 7 (a-c) and elaborated on this in the result section (line 362).

12. Is the statement regarding accumulation of more cells per generation meant to suggest that for each cell that enters division there are fewer cells remaining by division 4 or simply that there are fewer cells that made it to division 4. The former would suggest differential survival

In figure 7 we show more cells in each division. A combination of increased cell division and increased cell survival after each division result in a more expanded population. We have not measured survival, but we suspect that the increased abundance of IL-2 in dKO cultures would promote survival. We have not pursued this in detail, as the effects of IL-2 on T cell survival and proliferation *in vitro* do not correlate with their *in vivo* roles.

13. What happens to ZFP36 and ZFP36L1 expression in WT cells stimulated with anti-CD3 and CD28?

Previous work (<https://doi.org/10.7554/eLife.33057.001>) has shown that stimulation of CD4 or CD8 T cells with plate bound anti-CD3 and anti-CD28 induces the transient induction of the RBPs. Peptide stimulated CTLs show a similar transient kinetics. We have added this new data as SF3ab and commented on it in the results section (line 175). This data shows that upon antigen stimulation the expression of ZFP36 and ZFP36L1 peaks at around 3-4 hours after stimulation and by 24 hours the expression is reduced.

14. Why do the authors think the individual effector functions are differentially regulated at early versus late times (Fig.3)? It would be great to work this into the model.

ZFP36 and ZFP36L1 are expressed upon stimulation of naïve cells in a transient fashion. Both proteins are again re-expressed transiently upon antigen stimulation of CTLs.

In naïve T cells we think that ZFP36 and ZFP36L1 mainly control differentiation. This differentiation is linked to activation which is limited by the RBPs. The ZFPs coordinate multiple processes during the first 24 hours of T cell activation which involve the activation process, induction of differentiation and cell division and survival. It remains possible that the RBP also regulate epigenetic and metabolic processes. Thus, the ZFPs affect the early programming of T cells to become effector cells. The enhanced killing observed in CTLs when the RBPs are absent during initial activation and enhanced GranzymeB expression is a consequence of the activity of the RBPs in naïve cells. Thus, the mechanism for enhanced killing is installed early and is repressed by both RBPs.

Once differentiated to an effector cell, antigen also activates cellular processes that enable the cell to fulfil its role. Thus, ZFP36 and ZFP36L1 primarily regulate TNF and possibly other cytokines in CTL, but they do directly not limit cellular processes which result in enhanced killing in CTLs.

15. It would be helpful to discuss this pathway in the context of a normal immune response. This appears to be the brakes on the system to ensure the need for a stimulus that results in DC maturation. Some of this information is in the figure legend of SF6, but more in the body of the discussion would be useful.

We thank the reviewer for this insightful comment but we have not formally tested how the deletion of RBPs in T cells affects APCs. It is possible that the RBP enforce T cells to respond only when antigen is properly presented by APCs. We agree that this is an interesting issue and requires more scrutiny in the future in the context of autoimmunity. But at present we would not like to speculate even more on antigen presentation and autoimmunity and the role of the ZFPs in it.

Minor points:

1. It would be helpful to note in the text of the results the day p.i. or stimulation that the assay is performed as it is not consistent within the figures.

We would like to thank the reviewer to drawing our attention to this mistake. We have corrected the information in the figure legends which now correlates with the text.

2. Please add % numerical values to data in F3d and f.

We have added % numerical values to the data in Figure 3.

3. The authors should state “data not shown” after the finding where NFkB and IRF8 were measured after IL-2 was added.

This data as is shown in Figure 7 (h-k).

4. Delete “to” in first line of discussion.

We would like to keep the sentence as in the original version.

5. Please add reference for the Roquin paper described in Discussion.

We have substantially revised the discussion of this manuscript and no longer refer to Roquin.

6. Space needed “andTBET” in Discussion.

Thank you for pointing this error out to us. We have corrected accordingly.

7. The title for SF1 is incorrect as both flu and Lm are used. Consider dividing into two figures.

We restructured the figure legends and moved part of the data related to influenza into the main Figure 2.

8. Gamma symbol lost in legend of SF1.

We have corrected this

9. Please add the group explanations for I,II,III,IV, and V to legend of SF3.

We have added the explanations of the groups to legend of SF5.

Response to comments of Reviewer 2:

This study shows that the RNA binding proteins ZFP36 and ZFP36L1 control the extent to which naïve CD8+ T cells differentiate into effector cells that limit immunopathology and viral load, and eliminate antigen-loaded target cells in vivo. Mice doubly deficient (dKO) for these RBP lost less weight than WT mice after primary infection with the highly pathogenic PR8 strain of IAV, and were protected against lethal PR8 infection. Similar results were seen when WT mice were injected with small numbers of WT or dKO OT-I cells, demonstrating that effects were CD8+ T cell-intrinsic. By d10, the response of dKO cells was moderately skewed towards an SLEC phenotype, (although most of the surviving OT-I cells appeared to express neither CD127 nor KLRG-1). OT-I cells that could only express ZFP36 and ZFP36L1 during the first 24 hours after stimulation behaved like WT rather than dKO cells, consistent with the hypothesis that ZFP36 and ZFP36L1 act early in CD8+ T cell differentiation. Expression of transcription factors important for effector function was also enhanced in dKO cells. Finally, WT OT-I cells were significantly more dependent on CD28 costimulation compared with dKO OT-I cells.

Based on these results, the authors discuss how ZFP36 and ZFP36L1 limit CD8+ T cell activation and differentiation into effector cells. Rather than just limiting effector cell development, however, ZFP36 and ZFP36L1 might be critically important for the differentiation of naïve CD8+ T cells into MPEC rather than SLEC. Response skewed towards SLEC would be highly effective at viral clearance and limiting

immunopathology during primary infection, but at the expense of memory formation and protection against viral challenge. Do dKO OT-I cells make significantly impaired memory responses compared with WT OT-I cells to a second IAV infection? This would reframe the discussion away from limiting effector differentiation, towards an important role in memory formation.

Thank you very much for your careful reading of the manuscript and insightful comments. The point raised regarding memory formation in the absence of ZFP36 and ZFP36L1 is a very interesting and substantial question: Is the more efficacious and potent effector response happening at the expense of memory formation? Although we lack a detailed systematic body of results, we observe a reduced accumulation of MPEC which have a greater potential to form central memory. We have added this data on MPEC formation in influenza and Listeria infection (Figure 2c; SF2c) and described the results (line 150,164).

As the expansion of the effector pool at the peak of the response is reduced, we anticipate that the numbers of memory cells formed by dKO CD8 T cells will be reduced. What will require significant additional investigation is whether memory is qualitatively affected- taking into consideration the varieties of memory cell types - and their capacity for enhanced performance and protection upon reinfection. We are embarking on these studies, but want to generate a significant body of data that will be publishable as a complete study in its own right.

One intriguing result: adoptive transfer of 200 dKO OT-I cells had a dominant effect on the endogenous CD8+ T cell response, as shown in Fig.S1a,b. Transfer of 200 OT-I cells should result in engraftment of about 20 OT-I cells (assuming ~10% engraftment success rate), rather fewer than the number of naïve endogenous SIINFEKL-specific CD8+ T cells (70 - 600; reviewed in Jenkins et al, 2012, J. Immunol. 188: 4135). Fig. S1a, b, show similar numbers of endogenous SIINFEKL-specific CD8+ T cells and WT OT-I cells in the spleen and lung 10 days after infection with IAV. In contrast, the number of endogenous cells is reduced almost 10-fold when dKO OT-I cells were transferred. Perhaps these data could be incorporated into Fig. 2? They're certainly consistent with the idea of a rapid effector response by dKO cells that could reduce viral infection loads, as mentioned in the discussion.

As suggested, we have moved the data (previously SF1a), into the main Figure 2b. We have to apologize for a mistake in the former figure S1a and b. The previous figure the donor and recipient cells were mislabelled in the left panel (Former S1a). We now present a corrected version of the figure in which the open shaded circles represent donor cell numbers (WT or dKO), while the filled circles represent cells originating from the recipient. This data shows that we find reduced numbers of transferred cells when they are deficient for ZFP36 and ZFP36L1.

There's some concern about the very exciting culture conditions used to generate CTL in vitro (Fig. 3). CD8+ T cells expand extremely rapidly after SIINFEKL activation of OT-I spleen cells, differentiate into highly potent CTL even in the absence of IL-12, and must be diluted very frequently to maintain optimal viability. Cells grown in very high dose IL-2 (20nM = ~300ng/mL) without IL-7 or IL-15 generally don't revert to resting cells, are highly cytotoxic, and are extremely susceptible to apoptosis. The limited GzmB expression and relatively poor cytotoxicity (less than 50% lysis at 1:1 E:T) of WT OT-I cells after 7 days culture was surprising. More information on culture conditions and cell viability would be helpful.

We have added a more detailed description in the Materials and Methods section of the generation of CTLs *in vitro* (line 598). We analyzed the killing after 3hours which provided the

best dynamic range of the assay in our hands. At later timepoints (24h) all target cells were killed.

Minor points:

The diagram in Fig. 2a suggests that ZFP36 and ZFP36L1 are also expressed during the 3h in vitro cytotoxicity assay - has this been shown?

Yes. Upon restimulation of CTLs with peptide, both RBPs are expressed in a rapid and transient fashion. We have added Western blot data showing the expression of ZFP36 and ZFP36L1 by CTL upon antigen restimulation (SF3ab). We comment on this in the results section (line 176). See also the response to comment 13 of reviewer 1.

Could the flow data shown in Fig. 2b-f could be shown as un-smoothed pseudocolor plots?

Yes. We have changed the plots in Figure 2 into unsmoothed pseudocolor format.

Figures showing gating strategies and appropriate controls for cytokine secretion assays (see Figs. 2b-f; Fig 3 d,f; Fig. 6g) should be included in the Supplementary Data (Fig. 5 looks fine). This is especailly important for cells grown for

We have added the gating strategies for Figure 2 and Figure 3. to SF1c-e, SF4a.

Since granzyme B can be stored in granules, it would be very helpful to include intracellular cytokine staining of unstimulated as well as peptide-restimulated cells in Fig. 3d, or show these as supplementary figures.

We have included data on GranzymeB expression in CTLs prior to antigen/target cell encounter. We find that only CTLs where ZFP36 and ZFP36L1 were absent during initial stimulation have increased GranzymeB staining. We added this data into SF4d and mentioned the results in the text (line 205).

The data in Fig. 4a were presumably generated from peptide-stimulated naïve OT-I cells, not CTL as stated (p7, second paragraph, line 9).

The CLIP data is a compendium from two sources: One using a panZFP36 family antibody on CD4 T cells activated for 4 hours with CD3 and CD28. This is data that is reused from Moore et al (<https://www.ncbi.nlm.nih.gov/pmc/articles/PMC6033538/>). The ZFP36L1 iCLIP data is new data generated from peptide stimulated CTLs as stated. Thus, it is formally possible that there are specific targets in activated naive CD8 T cells that have been missed by this approach. However, there is a substantial overlap between the transcriptomes of these T cells sufficient to identify targets that can be further validated at the protein level. We have clarified this in the results section lines 225.

The data points for cells stimulated with CD3 without CD28 (mentioned in the text on p11) should be added to Fig 7b.

We have included the anti-CD3 only (no added anti-CD28) data point to Figure 7e.

Response to comments of Reviewer 3:

In this report Petkau et al. focuses on the role(s) of RNA binding proteins (RBPs) Zfp36 and Zfp36l1 during naive CD8 T cell activation and the response to Flu infection in mice. Analysis of mice in which genes encoding Zfp36 or Zfp36l1 were conditionally inactivated individually or in conjunction before naive T cells develop using the Cd4-Cre transgene (deletes in double-positive thymocytes) showed that elimination of both genes (double knockout [dKO]), but not either gene alone, increased viral clearance and decreased morbidity (less weight loss). Improved viral control in dKO mice is associated with increased frequencies of cells that expressed an effector cell phenotype (Klrg1hi, and increased Gzmb, IFNg, TNF expression) and increased in vivo killing function, and appears to be CD8 T cell intrinsic. The requirements for these RBPs appear to be early after TCR-activation because their inactivation after in vitro activation of naive CD8 T cells using CRISPR-Cas methods exhibited a less profound phenotype compared to Cre-mediated disruption. Multiple mRNA targets were identified using HITS-CLIP approach, and comparisons to mRNAs whose expression changes following initial activation of naive CD8 T cells. The protein-level expression of several of these targets (Il2, Nfkb2, Rel, Irf8 and Notch1) were found to be increased in cells lacking Zfp36 and Zfp36l1. The authors' main overarching conclusion is that the Zfp36 family of RNA binding proteins limit the speed and quality of the CD8 T cell response.

The molecular mechanisms that are involved in the initial activation and differentiation of naive CD8 T cells is a significant and interesting topic, and clearly have relevance for understanding how to control viral infections. The HITS-CLIP dataset in recently stimulated naive CD8 T cells appears to be high quality and will be a resource that should be of interest to the community. Overall, the manuscript is written clearly and the quality of the experiments and their presentation is high. However, a several issues limit the strength of the biological and mechanistic insights that can be currently drawn.

Thank you very much for your careful and insightful reading of the manuscript to which we reply in detail below.

Major points:

1. The finding in this study that Zfp36/Zfp36l1 also enhances antiviral immunity, in this case against influenza A infection, is important, but has less general impact because previous studies have reported have reported that Zfp36-deficiency enhances clearance of viral infection in other models (Moore et al. Elife, 2018).

We emphasise here and have done so in the introduction that the findings of Moore et al - (<https://www.ncbi.nlm.nih.gov/pmc/articles/PMC6033538/>) which show that enhanced immunity of ZFP36 KO mice to LCMV infection - is mediated by actions of ZFP36 in non-T cells.

Moore et al used two models: First the *Zfp36* germline knockout in which ZFP36 is absent in all cells of the mouse and in which viral clearance was enhanced; the second, a mixed bone marrow chimaera model in which all haematopoietic compartments are a mixture of wildtype cells and *Zfp36*-deficient cells. This identifies the cell-autonomous properties of *Zfp36*-deficient haematopoietic cells and Moore et al state that “*the kinetics of WT and KO T cell response in these animals upon LCMV infection were indistinguishable (Figure 7—figure supplement 3C–D).*” and they clarify further that “*in a mixed environment in vivo, Zfp36 KO and WT T cells show similar kinetics upon viral challenge*”. We advance the field by identifying

a clear cell-autonomous role for the RBPs specifically in CD8 T cells and show that heightened immunity is mediated by the absence of the RBPs in these cells.

2. This report provides some evidence that *Zfp36/Zfp36l1* regulates *Il2* mRNA, which is a different observation than previous studies. However, the extent of the mechanistic insight as to how this occurs limits the interpretations that can be drawn at present. The authors postulate that *Il2* dysregulation in the absence of *Zfp36/Zfp36l1* results from direct (*Il2* mRNA binding) and indirect (*Nfkb* mRNA binding) because of NF- κ B's appreciated role in promoting *Il2* transcription. However, the relative contribution of NF- κ B and direct action of *Zfp36/Zfp36l1* on *IL-2* message is not delineated and whether one or both explains the difference in *IL-2* expression between WT and dKO cells is presumed, but not demonstrated. Is *Il2* transcription affected, or is *IL-2* mRNA half-life different, and do these changes depend on the cis-elements in *Nfkb2* and *Il2* mRNAs in the absence/presence of *Zfp36/Zfp36l1*? Are there differences in NF- κ B-controlled regulatory elements in the *Il2* locus in WT versus dKO cells? Can these changes be linked to the *in vivo* phenotype of the dKO CD8 T cells during infection?

Previous work by other groups, with each using different experimental systems, has provided unclear and conflicting information on the regulation *IL2* by ZFP36.

The first from 2005 (<https://pubmed.ncbi.nlm.nih.gov/15634918/>) shows that splenocytes or purified T cells from *Zfp36*^{-/-} mice stimulated with anti-CD3 or anti-CD3+CD28 contain more *Il2* mRNA and release a greater amount of *IL-2* into the supernatant than control cells. This is correlated to an increase in *Il2* mRNA stability in *Zfp36*^{-/-} splenocytes. Because splenocytes contain a mixture of lymphoid and non-lymphoid cells and purified T cells include naïve and memory CD4 and CD8 T cells and NKT cells all of which can produce *IL-2*- it is unclear what cells were producing *IL-2*; the findings could be due to increased frequencies amongst *Zfp36* KO splenocytes or T cells of cell types that produce more *IL-2*.

The second study by Wang et al in 2017 (<https://www.nature.com/articles/s41467-017-00892-y>) found no differences in *IL-2* production in CD8 T cells that lack *Zfp36*. In this study the investigators stimulated total splenocytes with PMA and Ionomycin including Golgi Stop to detect cytokine production in CD8 T cells by intracellular flow cytometry.

The third study by Moore et al did not find increased *Il2* mRNA in *Zfp36* KO CD4 T cells (<https://www.ncbi.nlm.nih.gov/pmc/articles/PMC6033538/>). In fact, when we reanalysed their data, we found that in both the RNAseq as well as Riboseq data that *Il2* mRNA is, if changed at all, slightly less abundant in *Zfp36* KO CD4 T cells compared to WT controls. We have plotted the data in response **Figure 1.**

Response Figure 1: **a)** A comparison of *Ii2* mRNA in control and *Zfp36* KO CD4 T cells activated for either 4 hours with anti CD3 and CD28 or reactivated after 72 hours with PMA/Ionomycin. Error Bars indicate the adjusted p value. **b)** Ribo seq data shows the relative association of *Ii2* mRNA with ribosomes in the absence of ZFP36 relative to WT control CD4 T cells activated for 3 hours with anti CD3 and CD28. Error bars indicate the adjusted p value.

We have sought to clarify the regulation of *Ii2* in CD8 T cells and added new data (Figure 5e-g) which resolves the relative contributions of transcription and stability to the increased abundance of IL-2 that we reported.

Thus, our study unambiguously shows the regulation of IL2 by ZFP36 and ZFP36L1 in CD8 T cells. First, we show that the dKO OT-I cells secrete more IL-2 when stimulated with N4 peptide as compared to WT OT-I controls (Figure 5a and b). We provide new data that shows *Ii2* mRNA to be more abundant in the dKO especially at three- and six-hours following activation (Figure 5e). We next assessed whether absence of ZFP36 and ZFP36L1 affects the stability of *Ii2* mRNA. We used Triptolide to inhibit transcription in WT and dKO naive OT-I cells and measured the decay of the *Ii2* mRNA present at three-hours after stimulation. We found a clear increase in stability of *Ii2* mRNA in the absence of ZFP36 and ZFP36L1 (Figure 5f). Thus, ZFP36 and ZFP36L1 promote destabilization of *Ii2* mRNA after T cell activation. Previous studies analyzed the role of ZFP36 in the regulation of IL2, but did not assess the role of ZFP36L1.

We would also like to clarify that we find both ZFP36 and ZFP36L1 to independently suppress the production of IL2 by CD8 T cells. The deletion of both RBPs leads to an additive effect on the increased production of IL2 (Response figure 2). We do not think it helpful to incorporate this data into the manuscript because we have not scrutinised the phenotype of the single knockouts in detail.

Response Figure 2: IL2 secretion assay in ZFP36 and ZFP36L1 single and dKO. Naive OT-I CD8 T cells from WT, ZFP36KO, ZFP36L1KO and dKO were stimulated with 10^{-10} M N4

peptide for indicated timepoints and IL2 secretion was measured by IL2 cytokine secretion assay. Each data point represents a biological replicate.

In our study we report that the RBPs bind to mRNA encoding members of the NF- κ B pathway, *Nfkb1*, *Nfkb2* and *Rel*. Since NF- κ B has been demonstrated to be essential to produce IL-2 in T cells we speculated that the elevated levels of NF- κ B proteins which we find in dKO CD8 T cells shortly after activation contribute to an increased transcription of *Il2* mRNA. Therefore, we assayed *Il2* transcription in WT and dKO cells by measuring the presence of unspliced *Il2* mRNA which still retains intronic sequences. The levels of unspliced mRNA represent mRNA which has been transcribed "more recently" thus allowing us to get an indication of *Il2* transcription. Our results show no differences in unspliced mRNA between WT and dKO CD8 T cells (Figure 5g). Thus, we conclude that the increased *Il2* mRNA in the dKO CD8 T cells is mainly caused by the increased stability of the mRNA. We have added this new data into Figure 5e, f, g.

To test if increased IL2 is relevant to the cell-autonomous phenotype of dKO CD8 T cells in an infection we performed an experiment to test whether IL-2 produced by the dKO OT-I cells is necessary to accelerate the formation of effector cells. We deleted the *Il2* gene in dKO OT-I cells by nucleoporating Cas9/sgRNA Ribonucleoproteins targeting *Il2* and transferred these cells together with WT OT-I into CD45.1 recipients, then infected them with attLm-Ova. Both *Il2*-sufficient and *Il2*--deficient dKO cells yield a higher frequency of KLRG1⁺ cells in blood of recipients on day 5 post infection (SF6b-e). These results indicate that the enhanced effector phenotype is not due to IL-2 produced. We have described the new data in the text from line 285-311.

We show the RBPs repress further genes which regulate effector differentiation independently of IL2, including NF- κ B2, IRF8 and NOTCH1, and further transcription factors we identified as targets but have not validated in this manuscript like ZEB2 or MYC.

3. The nature of the interpretations in this study turn on the idea that *Zfp36/Zfp36l1* are mediating their functions following TCR stimulation of naive cells. However, naive cells from the dKO mice do not seem equivalent to WT cells before activation. NF- κ B2 and cRel protein expression both appear to be greater in dKO compared to WT cells before activation (Fig. 6e and f). Are there any differences in the phenotypes of cells isolated from naive WT and dKO mice before activation? A careful comparison of T cells from WT and dKO mice is important to demonstrate if cells from both backgrounds are on equal footing before activation, or to at least document what is already different before initial activation.

We agree with the reviewer that it is formally possible that "naïve" CD8 T cells from CD4cre *Zfp36/Zfp36l1* mice may have different properties from WT naive cells. However, we did not observe any developmental abnormalities of CD8 T cells in the thymus of non-transgenic or OT1 transgenic mice. The numbers of peripheral cells with a naïve cell-surface (CD44^{low}/CD62L^{high}) phenotype is 50% less and the staining of peripheral T cells with a panel of antibodies recognising different TCRV β revealed no skewing for the repertoire. We acknowledge this as a limitation of our current study.

4. The authors' interpretation that the improved clearance of flu infection in mice lacking *Zfp36/Zfp36l1* is due to enhanced effector functions in responding CD8 T cells. The kinetics of the infection are different in WT mice compared to the dKO mice, which undoubtedly will have affected the number and phenotype of the responding cells. This makes apples-to-apples comparisons between different groups of mice challenging,

and some of the observed differences in the frequencies of phenotypic populations of CD8 T cells could relate to differences in the infection kinetics in WT versus dKO mice.

We see evidence, early in the response, that is consistent with the kinetics of infection being similar in both genotypes because the number of T cells and viral loads are the same. Later in the response we see a 50% reduction in CTL lacking the RBP and we speculate this to be due, in part, to an earlier onset of the contraction phase that follows reduced amounts of antigen. Please also note our response to question no 1 of reviewer 1.

In addition, to what extent the differences in the flu response are due to altered effector function capability, or differences in effector cell accumulation in the absence of Zfp36/Zfp36l1 is not clearly resolved. The data presented regarding effector cell numbers in infected mice is confusing. Supplemental Fig 1a and 1b appears to be contradictory with the text. The figure indicates that there is a reduction in Ova-specific “recipient” cells, while the number of “donor” (OT-I) cells is largely unchanged between WT and dKO groups. However, the text states that: “Consistent with our studies in intact mice there are fewer dKO OT-I cells in the lung and the spleen at day 10 of infection than in mice with transferred WT OT-I cells (Supplementary figure 1a, b)”.

We appreciate the care with which the reviewer has scrutinised the data and would like to apologize for causing confusion about the recovery of transferred OT-I cells and antigen specific host cells. This is due to a mistake in the labelling of symbols in former SF1a. This figure has been corrected and was now moved to Figure 2b (Please also see response to question no 2 of reviewer 2).

Furthermore, it is difficult to reconcile the conclusion that there are fewer effector cells in dKO mice compared to WT mice, but there is increased target cell killing in dKO recipients during the in vivo lysis assay (Fig 1a), because the killing of targets in vivo is thought to proceed according to mass action, and even memory cells that have virtually undetectable perforin/granzyme B protein and killing activity in vitro (i.e., central memory cells) kill almost as rapidly as effector cells from the peak LCMV response when assayed in vivo (Barber et al. 2003, J. Immunol).

We find the numbers of effectors cells are reduced by 50% at day ten. Our *in vitro* and *in vivo* killing assays demonstrate that the dKO CTLs have a greater killing potential on a per cell basis. Here, we are not making any claims regarding the mechanism of killing. Our observation of increased expression of GranzymeB in dKO CTLs serves as a description of a characteristic in these cells which is altered compared to WT CTLs. It is well established that the expression of GranzymeB and Perforin are essential for CTLs to exert their cytotoxic functions, thus it correlates with the enhanced cell killing per cell by the dKO CTLs. Also, for memory cell killing Perforin and GranzymeB, which are rapidly induced upon activation (as also suspected by Barber et al), are essential. However, we cannot resolve here whether the increased amount of GranzymeB we observe in dKO CTLs is sufficient to explain the enhanced killing. We believe that multiple cellular processes are involved in successful killing, including cytoskeletal remodelling, refilling of cytotoxic granules and metabolic properties. We also believe that it is entirely feasible that fewer but more effective CTLs can clear more infected cells.

Altogether, these issues bring into question whether enhanced effector functions in dKO cells is the explanation of the improved viral control, or if it could relate to more rapid accumulation of effector cells. The authors’ observation that Zfp36/Zfp36l1 bound

to the key cell cycle regulators Cdk1 and Cdk6 could be an indication that the rate at which responding cells enter the response is accelerated in the absence of Zfp36/Zfp36l1, but this is not explored in terms of cell numbers early in the response.

We show a more rapid accumulation of effectors and that these effectors are more potent. We do not distinguish whether either of these by itself is sufficient to account for enhanced immunity. We have included new data that T cell numbers early in the response are not different (new figure 1 f and g). It is not possible for us to analyse the proliferation of these cells at the first days of the response due to very low numbers *in vivo*, thus we do not know whether the dKO cells enter cell cycle earlier as we observe *in vitro*. We agree that cell cycle regulators like Cdk1 and Cdk6 are part of the machinery controlling T cell activation and differentiation are likely suppressed by ZFP36 and ZFP36L1.

Specific comments/suggestions:

1. This sentence in the results is confusing: “To identify direct targets of the RBP we performed ZFP36L1 “individual-nucleotide resolution UV crosslinking and immunoprecipitation” (iCLIP) on OT-I CTLs stimulated with peptide for three hours and combined this with “high-throughput sequencing of RNA isolated by crosslinking immunoprecipitation” (HITS-CLIP) from CD4 T cells activated for 4 hours”. What was actually done here?

To identify candidate RNAs bound by the RBPs we used two datasets: iCLIP from ZFP36L1 from peptide stimulated CTL generated in our lab and pan-ZFP36 family antibody HITS-CLIP from naïve mouse CD4 T cells were activated with plate bound CD3 and CD28 generated by Moore et al. (<https://www.ncbi.nlm.nih.gov/pmc/articles/PMC6033538/>). The merged data sets identify a unified list of target RNAs of these RBPs, which includes common targets of both RBPs and also genes which were identified as targets exclusively for ZFP36 and ZFP36L1. We have clarified this in the manuscript on line 225. Please also see our response to question “minor points 6” of reviewer no 2.

2. The rationale for the selection of experiments in Fig. 4 seems incomplete. An important dataset that seems to be missing is one in which genes whose expression is altered in the absence of Zfp36/Zfp36l1. Do these RNA binding proteins universally negatively regulate the half-life of their bound targets, or are there different modes of action depending on which mRNAs or which elements in those RNAs they bind?

We acknowledge the limitations of our approach, but are firmly of the view that the datasets that we have made use of have provided a significant level of new mechanistic insight into pathways and processes relevant to CD8 cell fates.

Future studies can examine, using a multi-omics approach, the impact of the mutations on the kinetics of RNA and protein amounts and turnover and correlate this with a kinetic of RNA-protein interactions. Such a substantial body of work will provide more details, and will certainly resolve the identities of direct and indirect targets amongst differently expressed genes and help to resolve whether these targets are regulated via RNA decay and/or translational repression. But such an approach is not guaranteed to provide a deeper level of understanding into the biology of differentiation than provided in this manuscript.

3. Ideally, another Cre-driver that acts after the thymus seems to be an important complement to the study. In addition, all comparisons in the study use mice and cells with Zfp36/Zfp36l1 “floxed” alleles without CD4-Cre transgene as the negative control. A formal comparison to CD4-Cre+ cells/mice that have WT Zfp36/Zfp36l1 alleles is also

an important negative control, because Cre can have deleterious effects, especially in thymocytes.

The suggestion of the reviewer to use a post-thymic system of cre-mediated deletion is entirely appropriate for future studies which build on this work.

For the infection studies we are strong advocates of the use of littermate controls, in preference to mice from other colonies, because of the confounding effects arising from the lack of cohousing prior to and after weaning.

The reviewer raises the important question of Cre toxicity which has been found for many cre-transgenic lines when investigators have bothered to look. The allele we are using has been well-characterised in this respect: mice homozygous for the CD4cre BAC transgene ([10.1371/journal.pone.0046590](https://doi.org/10.1371/journal.pone.0046590)) do show a small reduction in thymic cellularity but no detectable effect on the CD4/CD8 distribution. However, the same study shows no detectable effect when the transgene is hemizygous. In all of our experiments we use mice which are hemizygous. An independent study (<https://pubmed.ncbi.nlm.nih.gov/27345256/>) confirms this.

Moreover, the majority of our experiments in this manuscript are performed with mature post-thymic CD8 T cells. These cells do not express the cre-transgene, an observation we have included in figure SF1a, b. We refer to this observation in line 139 of the results section. Furthermore, we find throughout our experiments a gain of function which we suggest is not compatible with a toxic effect of cre.

4. Figure 4b. What does the red dotted line indicate?

The dotted line indicates the log₂ FC=1.3 cut off used to identify the most strongly regulated genes. We added this explanation to the figure legends.

5. Are there differences in IL-2 regulated genes in Zfp36/Zfp3611-deficient CD8 T cells as a consequence of increased or accelerated IL-2 expression?

Some genes which drive effector differentiation and function including Tbet and Blimp1 are targets of IL-2. However, we suspect that IL-2 will not have a dominant role in driving effector differentiation and function since *in vivo* there are many different inflammatory cytokines which will contribute. Our experiments which we mention in response to question no 2 show that increased autocrine IL2 in dKO cells is not necessary to drive accelerated SLEC formation.

Thus overall, we think that the ZFP36-family limit the differentiation process at multiple levels, which all contribute to the observed phenotype. First, we see that the ZFPs limit the production of key cytokines in naïve CD8 T cells and, despite much research, it remains hard to estimate what the contribution of each of these cytokines is. Secondly, the ZFPs also directly regulate transcription factors as we have shown here. Some of these are linked to cytokine production, but others - IRF8 and Notch1 - are not necessarily linked to IL2 and TNF production. Third, our omics analysis is consistent with a hypothesis that the ZFPs also regulate metabolic and cell cycle processes which we did not extensively investigate here but plausibly contribute to cell fate decisions by CD8 T cells.

6. More clearly delineating in the text the situations in which more rapid gene expression kinetics versus qualitative changes in gene expression are evident would provide additional clarity to readers.

We are unsure what the reviewer has in mind when they refer to qualitative changes in gene expression. We interpret qualitative changes in gene expression to refer to alternative 5' or 3'

end formation or splicing which are factors that we have not taken into consideration for this study. However, we tried to conceptualize our idea of how the RBPs act on consequential suppression of their target genes in a temporally resolved fashion in our new graphical abstract in SF8.

REVIEWER COMMENTS

Reviewer #1 (Remarks to the Author):

The authors have added new data that significantly improves the manuscript and addresses the concerns raised. The reviewer would like to clarify that for comment 15-no speculation of RBPs in DC was intended, simply that this pathway in T cells may help ensure the need that DC have undergone maturation.

Reviewer #2 (Remarks to the Author):

The authors have made substantial revisions to their manuscript that have addressed all the points I raised, and many of the concerns raised by the other two reviewers.

It was helpful to see additional information on MPEC vs SLEC numbers. The authors agree that effects on memory cell generation are important areas for future study. The more detailed description of the in vitro methods used to generate CTL is appreciated; it does reinforce some concern about these very exciting culture conditions, but doesn't really affect interpretation of the results.

One very minor point: the revised Figure 7e shows data points on the Y-axis which may represent the "anti-CD3 only" values, but the Y-axis label conflicts with this, and should be corrected.

Reviewer #3 (Remarks to the Author):

The authors have addressed in a thorough manner the most important questions that my review raised, especially with regard to regulation of IL-2 expression. Very nice work.

1. The authors make a strong conclusion that the "tempo" of CD8 T cell differentiation is increased in cells lacking Zfp36/Zfp3611 proteins. In this regard, I would ask the authors to consider two issues that they might address in their text:

(a) It appears that Zfp36/Zfp3611 are instrumental in providing "negative feedback" within the first few hours of naive CD8 T cell activation. Although the response of Zfp36/Zfp3611-deficient CD8 T cells exhibits more efficient control of Flu infection, and the T cell response shows a slightly depressed kinetic in cell accumulation, I am not sure the data indicate that the differentiation of the CD8 T cells is necessarily accelerated. Rather, it seems the data indicate that the duration with which the responding cells initially express molecules that drive effector cell gene expression/capabilities is increased in magnitude above normal (with relatively similar initial kinetics, Fig. 5B and 5C show this nicely) and therefore their expression becomes protracted rather than transient, compared to when the negative feedback checkpoints are in place. The net result in vivo appears to be truncation of the infection, which removes the antigenic stimulus "prematurely" compared to the wildtype scenario. In short, do the authors think that differentiation is accelerated on a per cell basis without Zfp36/Zfp3611, or is the population of activated cells more synchronized from the start, resulting in a greater frequency of cytotoxic KLRG1-hi cells relative to what normally occurs with a population of wildtype cells?

(b) The authors acknowledge that naive CD8 T cells lacking Zfp36/Zfp3611 have some alterations compared to wildtype naive cells. It would be helpful to make sure this is not lost on the readers and that they are made clear about these differences, and how this could limit/contribute the interpretations.

2. I would consider a little more flexibility in the conclusions regarding whether or not IL-2 contributes to the altered pattern of differentiation of cells lacking Zfp36/Zfp3611, because it is possible that the RNP-Cas9-mediated disruption of IL-2 could have come too late to have sufficiently hampered the IL-

2-mediated effects, especially given that the authors show Zfp36/Zfp36l1 control the half-life of pre-formed IL2 message which is induced rapidly upon initial TCR stimulation. It seems that the IL-2-disrupted group did appear to trend having lower frequency of KLRG1-hi cells (Supp Fig. 5B), even if the means were not statistically different at a stringent confidence level for the relatively few mice in the experiment. Also, it was not shown how quickly and completely IL2 was disrupted after nucleofection of the naive cells, as only the day 7 data were shown. Going easy on this conclusion might be warranted.

It seems like it would be reasonable if the discussion could include consideration of some of these issues.

Reviewer #1 (Remarks to the Author):

The authors have added new data that significantly improves the manuscript and addresses the concerns raised. The reviewer would like to clarify that for comment 15-no speculation of RBPs in DC was intended, simply that this pathway in T cells may help ensure the need that DC have undergone maturation.

We thank the reviewer for their thorough review of our manuscript. We agree with the reviewer that the pathway regulated by the RBPs may be part of the mechanisms that enforce ensures T cells respond only once antigen is presented by “matured” APCs. We have added a statement to this effect in our discussion.

Reviewer #2 (Remarks to the Author):

The authors have made substantial revisions to their manuscript that have addressed all the points I raised, and many of the concerns raised by the other two reviewers.

It was helpful to see additional information on MPEC vs SLEC numbers. The authors agree that effects on memory cell generation are important areas for future study. The more detailed description of the in vitro methods used to generate CTL is appreciated; it does reinforce some concern about these very exciting culture conditions, but doesn't really affect interpretation of the results.

One very minor point: the revised Figure 7e shows data points on the Y-axis which may represent the "anti-CD3 only" values, but the Y-axis label conflicts with this, and should be corrected.

We would like to thank the reviewer for their comments on our manuscript and we are looking forward to investigate the memory responses in the absence of RBPs.

We have corrected the x-axis of Figure 7e.

Reviewer #3 (Remarks to the Author):

The authors have addressed in a thorough manner the most important questions that my review raised, especially with regard to regulation of IL-2 expression. Very nice work.

1. The authors make a strong conclusion that the “tempo” of CD8 T cell differentiation is increased in cells lacking Zfp36/Zfp36l1 proteins. In this regard, I would ask the authors to consider two issues that they might address in their text:

(a) It appears that Zfp36/Zfp36l1 are instrumental in providing “negative feedback” within the first few hours of naive CD8 T cell activation. Although the response of Zfp36/Zfp36l1-deficient CD8 T cells exhibits more efficient control of Flu infection, and the T cell response shows a slightly depressed kinetic in cell accumulation, I am not sure the data indicate that the differentiation of the

CD8 T cells is necessarily accelerated. Rather, it seems the data indicate that the duration with which the responding cells initially express molecules that drive effector cell gene expression/capabilities is increased in magnitude above normal (with relatively similar initial kinetics, Fig. 5B and 5C show this nicely) and therefore their expression becomes protracted rather than transient, compared to when the negative feedback checkpoints are in place. The net result in vivo appears to be truncation of the infection, which removes the antigenic stimulus “prematurely” compared to the wildtype scenario. In short, do the authors think that differentiation is accelerated on a per cell basis without Zfp36/Zfp3611, or is the population of activated cells more synchronized from the start, resulting in a greater frequency of cytotoxic KLRG1-hi cells relative to what normally occurs with a population of wildtype cells?

We observe the earlier acquisition of GranzymeB by CD8 T cells which lack RBPs in the influenza model and also during in vitro differentiation. Moreover, in the listeria model we observe a more rapid acquisition of the SLEC phenotype by CD8 T cells which lack RBPs prior to the peak of the T cell response in blood. These observations led us to conclude that effector differentiation is accelerated. On a population level we envision that during an infection, which is the sum of antigen dose and quality, and the inflammatory milieu, the dKO cells are more likely to be skewed to form SLEC at the population level. Thus, the dKO T cells have a lower overall threshold to respond to antigen and inflammation and a greater likelihood to acquire the SLEC phenotype. This is reflected in increased frequencies of SLEC during listeria and influenza infections. However, we also want to emphasize that our in vivo and in vitro cytotoxicity assays, the latter of which is performed with CTLs which have developed under highly polarizing conditions, show increased cytotoxicity in the absence of RBP on a per cell basis.

To sum this up there is a greater likelihood for dKO CD8 T cells to form a relatively greater SLEC population, while at the same time on an individual cellular level dKO CD8 T cells have increased cytotoxic potential.

We have tried to address the reviewer’s comments regarding protracted target expression, the accelerated effector differentiation and cessation of infection in our updated discussion. However we would not like to extend our discussion to such great detail as we have done in this response to the reviewer.

(b) The authors acknowledge that naive CD8 T cells lacking Zfp36/Zfp3611 have some alterations compared to wildtype naive cells. It would be helpful to make sure this is not lost on the readers and that they are made clear about these differences, and how this could limit/contribute the interpretations.

We have not noted any developmental abnormalities and also no obvious differences in T cell selection during thymic development in T cells which lack RBP. We do acknowledge in our updated discussion that due to the fact that the RBPs limit T cell activation it is possible that formally naïve CD8 T cells will have experienced self-antigen stronger than their WT counterparts. However we did not find any effector molecules or transcription factors associated with differentiation to be expressed in dKO naïve cells prior to activation. We have not studied potential effects on chromatin accessibility.

2. I would consider a little more flexibility in the conclusions regarding whether or not IL-2 contributes to the altered pattern of differentiation of cells lacking Zfp36/Zfp36l1, because it is possible that the RNP-Cas9-mediated disruption of IL-2 could have come too late to have sufficiently hampered the IL-2-mediated effects, especially given that the authors show Zfp36/Zfp36l1 control the half-life of pre-formed IL2 message which is induced rapidly upon initial TCR stimulation. It seems that the IL-2-disrupted group did appear to trend having lower frequency of KLRG1-hi cells (Supp Fig. 5B), even if the means were not statistically different at a stringent confidence level for the relatively few mice in the experiment. Also, it was not shown how quickly and completely IL2 was disrupted after nucleofection of the naive cells, as only the day 7 data were shown. Going easy on this conclusion might be warranted.

As suggested by the reviewer, in the revised version of the discussion, we have chosen a more careful formulation of our conclusion emphasizing that increased autocrine IL2 is likely not required for accelerated SLEC formation. We agree with the reviewer that we have not formally tested whether IL2 is already deleted in the first hours after activation in vivo. However we would like to stress that the deletion of the IL2 gene mediated by Cas9 is likely to happen in the first 24 hours after nucleoporation and thus well before stimulation with antigen in vivo and subsequent proliferation. This assertion is supported by evidence that in other cell types nucleoporated Cas9 is not detectable after 24 hours (DOI: [10.1101/gr.171322.113](https://doi.org/10.1101/gr.171322.113)). Together with this notion, since the absence of autocrine IL2 does not disadvantage CD8 T cell proliferation in vivo (DOI: [10.1038/nature04790](https://doi.org/10.1038/nature04790) , DOI: [10.1038/ni.2079](https://doi.org/10.1038/ni.2079)), the very low frequency of cells still able to produce IL2 on day 7 of the infection suggests that the initial deletion was very efficient.

It seems like it would be reasonable if the discussion could include consideration of some of these issues.

We would like to thank the reviewer for encouraging us to expand our discussion on these issues and we think that we have improved the discussion in this regard.